# Augmentation-Free Dense Contrastive Knowledge Distillation for Efficient Semantic Segmentation

**Jiawei Fan**[*]
Intel Labs China
jiawei.fan@intel.com

**Chao Li**
Intel Labs China
chao3.li@intel.com

**Xiaolong Liu**
HoloMatic Technology Co. Ltd.
liuxiaolong@holomatic.com

**Meina Song**
BUPT
mnsong@bupt.edu.cn

**Anbang Yao**[†]
Intel Labs China
anbang.yao@intel.com

## Abstract

In recent years, knowledge distillation methods based on contrastive learning have achieved promising results on image classification and object detection tasks. However, in this line of research, we note that less attention is paid to semantic segmentation. Existing methods heavily rely on data augmentation and memory buffer, which entail high computational resource demands when applying them to handle semantic segmentation that requires to preserve high-resolution feature maps for making dense pixel-wise predictions. In order to address this problem, we present Augmentation-free Dense Contrastive Knowledge Distillation (Af-DCD), a new contrastive distillation learning paradigm to train compact and accurate deep neural networks for semantic segmentation applications. Af-DCD leverages a masked feature mimicking strategy, and formulates a novel contrastive learning loss via taking advantage of tactful feature partitions across both channel and spatial dimensions, allowing to effectively transfer dense and structured local knowledge learnt by the teacher model to a target student model while maintaining training efficiency. Extensive experiments on five mainstream benchmarks with various teacher-student network pairs demonstrate the effectiveness of our approach. For instance, the DeepLabV3-Res18|DeepLabV3-MBV2 model trained by Af-DCD reaches 77.03%|76.38% mIOU on Cityscapes dataset when choosing DeepLabV3-Res101 as the teacher, setting new performance records. Besides that, Af-DCD achieves an absolute mIOU improvement of 3.26%|3.04%|2.75%|2.30%|1.42% compared with individually trained counterpart on Cityscapes|Pascal VOC|Camvid|ADE20K|COCO-Stuff-164K. Code is available at https://github.com/OSVAI/Af-DCD.

## 1 Introduction

In computer vision, semantic segmentation is a mainstream dense prediction task aiming to assign the corresponding class to every pixel of input images. Although fundamental models in this task have achieved remarkable progress [1–3], the heavy computational burden and latency of these models severely prohibit their deployment on resource-constrained devices. As a popular solution to this bottleneck, Knowledge Distillation (KD) [4] aims to transfer the knowledge from large models to light-weight ones. Considering KD in supervised semantic segmentation, teachers usually have

---

[*]This work was done when the first two authors were interns at Intel Labs China, supervised by Anbang Yao who proposed the original idea and led the project.

[†]Corresponding author.

37th Conference on Neural Information Processing Systems (NeurIPS 2023).

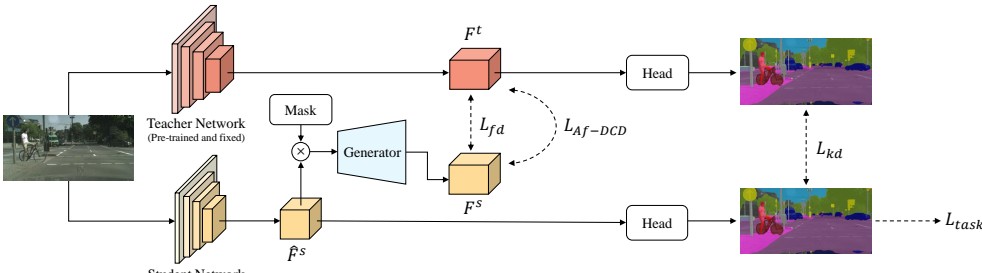

Figure 1: The overall framework of Af-DCD, which contains two major parts: (i) masked feature reconstruction; (ii) augmentation-free contrastive distillation loss. Detail illustrations on Af-DCD are shown in Figure 2.

stronger ability of capturing detailed local information of images where the *dense and structured knowledge* contributes significantly to segment complicated areas, such as boundary and occlusion. Therefore, distillation methods tailored to this task should keep the integrity of dense and structured knowledge, rather than compressing it excessively such as simply using global representations or salient areas. As a result, directly applying traditional KD methods in semantic segmentation, like vanilla KD [4] or feature imitation [5], cannot achieve promising results in most cases.

Thanks to the development of contrastive learning, some distillation methods, like SSKD [6], CRD [7] and G-DetKD [8], employed this paradigm into their designs, which promoted the performance significantly, as they exploited teacher's intrinsic knowledge in the consistencies among different augmentations or various same-category instances. In semantic segmentation, CIRKD [9] designed an implicit contrastive method which aimed to guarantee pixel-pixel and pixel-region consistencies between teacher and student by involving additional contrastive samples. On one side, these existing methods have achieved remarkable progress in image-level or object-level contrastive mimicking. On the other side, however, the efficacy of the aforementioned methods heavily relies on the augmentation and memory buffer, which entail heavy computational and memory cost. Besides, previous contrastive distillation methods have not explicitly modeled relations among pixel-wise or more fine-grained representations within each local patch to transfer teacher's dense and structured knowledge to student, which is critical in semantic segmentation. Thus, when designing a contrastive distillation method for this task, two vital technical problems should be tackled: **(i) High resource demands**: No matter leveraging augmented samples or storing feature maps in memory buffer, more computational or storage cost is required. For one, the forward of augmented samples incurs extra computational cost. For another, the output feature maps for semantic segmentation are in high resolution. If we want to make dense and structured contrasting, these feature maps should be originally preserved, which occupy a large amount of memories; **(ii) Structured knowledge transfer**: Although CIRKD defined pixel-wise alignment between student and teacher, it employed contrastive pixel-wise representations from other images, rather than local areas within the same image. It means that no explicit contrasting was defined between student-specific pixel-wise representation to each teacher's pixel-wise representation within local areas, which was agnostic to the structure in teacher's feature. *In brief, a contrastive distillation method specially tailored to semantic segmentation, which also entails no extra high resource demands (data augmentation and memory buffer), is essential.*

Driven by achieving this target, we first look into the aforementioned two problems and surprisingly discover both of them are incurred from the simple inheritance in the basic definitions of traditional contrastive learning [10, 11]. Therefore, our Augmentation-free Dense Contrastive Knowledge Distillation (Af-DCD) for supervised semantic segmentation tasks can come out, by re-defining these basic concepts: (i) *Contrasting samples* are not coarse-grained representations of images or objects. Instead, in our Af-DCD, we move further on pixel-level representations and divide each of them into several disjoint partitions (fine-grained representations), which are treated as our contrasting samples; (ii) *Positive and negative pairs* are not conditioned on categories or objects. Instead, based on the definition of contrasting samples, we define our pairs in a specific teacher-student feature pair $(F^t, F^s)$, where fine-grained representations having the same absolute positions in feature maps are used to formulate positive pairs, while representations having different absolute positions but within neighbourhoods are used as negative pairs. Based on these definitions, our Af-DCD can naturally tackle those two problems discussed above. To the first problem, unlike previous methods which had to introduce data augmentation and memory buffer to construct sufficient positive and negative pairs,

our Af-DCD can directly construct these pairs in each local patch without any extra computational and memory cost. To the second problem, we approach it by introducing two new contrastive designs, **Spatial Contrasting** and **Channel Contrasting**, which enable student to capture teacher's local dense and structured knowledge by focusing on contextual and positional channel-group information, respectively. As a natural progression, we introduce a hybrid design called **Omni-Contrasting**, which combines Spatial Contrasting and Channel Contrasting in a neat manner. This design facilitates the simultaneous transfer of both types of information from the teacher to the student, enabling effective augmentation-free contrastive knowledge distillation. Despite leveraging rather dense contrasting, our Af-DCD also performs efficiently. This is due to the patch separation technique we use, which significantly reduces the computational complexity. Additionally, the distance measuring and contrasting calculation can be carried out in parallel, further enhancing the overall efficiency of our method.

Experimental results demonstrate that: (i) Af-DCD exhibits superior performance compared to state-of-the-art methods, on various benchmarks with different teacher-student network pairs; (ii) Af-DCD exhibits even more significant improvements on larger datasets, such as ADE20K, indicating it can enhance student's generalization capability. Moreover, our analytical experiments illustrate that by effectively learning teacher's self-similarity distribution within neighbourhoods and thus reducing the fine-grained feature distances to the teacher, Af-DCD benefits addressing difficult scenarios in semantic segmentation.

## 2 Related Works

**Knowledge Distillation.** Knowledge distillation can be generally divided into probability-based approach and feature-based approach. Specifically, the former one forces student to mimic teacher's logits as soft-labels [4, 12], while the later one leverages teacher's hidden feature maps or its variants [5, 13, 14] as distillation supervisions. Some recent feature-based approaches [15–17] designed their distillation methods based on masked image modeling mechanism [18, 19] and achieved promising performance in various tasks, as the feature reconstruction can enhance the interdependencies among pixels, which is beneficial to feature distillation.

**Knowledge Distillation in Semantic Segmentation.** As semantic segmentation is a dense prediction task, the methods specially designed for this task aimed to capture teacher's structured local information. In order to achieve this target, SKD [20] directly measured the similarities of pixel-wise representations between teacher and student, while *He et al.* [21] leveraged non-local operation with autoencoder to encode local information. CWD [22] evaluated channel-wise pixel distribution contributing to learn teacher's spatial information in each individual channel. Some methods further explored intrinsic knowledge among different samples. For instance, IFVD [23] measured distances from prototypes of different classes and forced student to mimic teacher's intra-class relations, and CIRKD [9] forced student to keep pixel-wise and region-wise consistencies to teacher among various samples in memory buffer.

**Contrastive Knowledge Distillation.** Inspired by the development of contrastive learning in self-supervised [10, 24] and supervised tasks [11], some recent works leveraged this paradigm and designed new contrastive distillation methods, which made contrasting between teacher's and student's features by employing different views generated from data augmentation [6], or using various samples' features [25, 7] as well as gradients [26] stored in memory buffer. Specifically in dense prediction tasks, G-DetKD [8] constructed ROI feature pairs and executed soft semantic-guided matching which promoted the performance in object detection, and CIRKD [9] designed an implicit contrastive method which leveraged both pixel and region representations to learn structured information in spatial dimension.

## 3 Method

### 3.1 Overall Framework

Figure 1 illustrates our distillation process in semantic segmentation from a pre-trained teacher $T$ to a student $S$ which needs to be trained on a specific dataset. The output features of $T$ and $S$ are denoted as $\hat{F}^t \in \mathbb{R}^{H \times W \times C^t}$ and $\hat{F}^s \in \mathbb{R}^{H \times W \times C^s}$ respectively, where $H$ and $W$ are height and width of

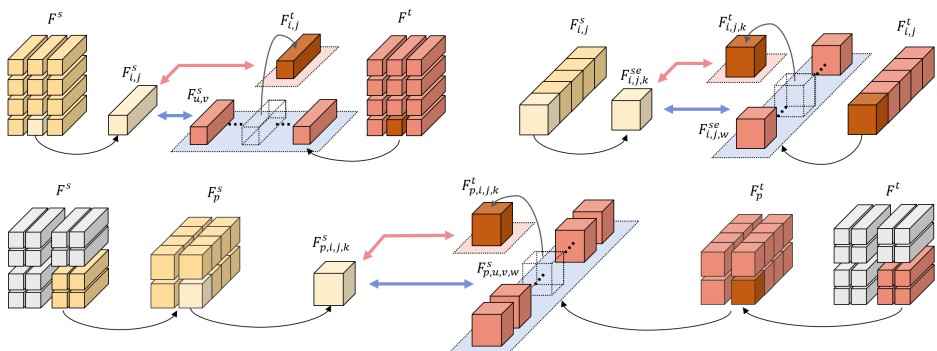

Figure 2: Detailed illustrations on three different types of Af-DCD, which are Spatial Contrasting (top left), Channel Contrasting (top right) and Omni-Contrasting (bottom). For brevity, the contrasting process is illustrated merely using a specific contrastive sample in student feature maps, denoted as $F_{i,j}^s$, $F_{i,j,k}^s$, $F_{p,i,j,k}^s$ in three Af-DCD designs, respectively. The red arrows denote constructing positive pairs, while the blue arrows denote constructing negative pairs. The gray blocks denote other patches which are not considered in calculating the loss in this patch.

feature maps, $C^t$ and $C^s$ are the number of channels of teacher and student, respectively. Our method uses a random mask $M \in \mathbb{R}^{H \times W \times 1}$ to overlap $\hat{F}^s$ in spatial dimension and then inputs the masked feature into a generator, which can be formulated as $F^s = f_{generator}(M \odot \hat{F}^s)$. Finally, we can get $F^s \in \mathbb{R}^{H \times W \times C^t}$, which has the same dimension as teacher's feature. In order to fit general definition, teacher's feature is also processed by a transform $F^t = f_t(\hat{F}^t)$. In our design, $f_t$ is an identity transformation, thus $F^t = \hat{F}^t$. After these operations, the reconstructed student feature $F^s$ has been projected into the same feature space as teacher's feature $F^t$. The feature imitation loss $L_{fd}$ and our Af-DCD loss $L_{Af-DCD}$ are calculated based on feature pair $(F^t, F^s)$. Besides, following the settings in [9, 22], we also employ vanilla logits-based KD loss [4]. In general, the overall loss of our method can be defined as:

$$L = L_{task} + \lambda_1 L_{kd} + \lambda_2 L_{fd} + \lambda_3 L_{Af-DCD}, \tag{1}$$

where $\lambda_i$ is the balancing weight.

### 3.2 Augmentation-free Dense Contrastive Knowledge Distillation

After illustrating the overall framework, we further explain our core design in detail, which contains two major parts, $L_{fd}$ and $L_{Af-DCD}$. As we mentioned above, both of them are based on the same feature pair $(F^t, F^s)$. Therefore, in order to clarify our design, we first answer two key questions: (i) What are the targets of $L_{fd}$ and $L_{Af-DCD}$ separately? (ii) How these two losses cooperate with each other? To the first question, $L_{fd}$ directly help student to imitate teacher's global and salient features, while $L_{Af-DCD}$ is designed to mimic teacher's dense structured knowledge within local areas, which can further force $F^s$ to approach $F^t$ in the micro and fine-grained views. To the second question, in terms of mimicking dense and structured knowledge, these two losses are inextricably bound up with each other. For one, $L_{fd}$ provides $L_{Af-DCD}$ an absolute imitation trend, which cannot be implemented in contrastive loss. For another, $L_{Af-DCD}$ offers $L_{fd}$ essential constrains on encouraging this process towards mimicking teacher's structured knowledge in each local patch, making student further approach teacher.

**Masked Reconstruction based Feature Imitation.** The feature imitation loss $L_{fd}$ is a basic part of our design, which forces student feature $F^s$ to imitate teacher feature $F^t$ directly. Previous methods commonly leverage simple upsampling to align teacher's and student's feature dimensions. Recently, some methods leveraged masked image reconstruction [15, 17], which is a stronger baseline, as it can promote the performance by strengthening the interdependencies among pixels. This process can be formulated as:

$$F^s = f_{generator}(M \odot \hat{F}^s), \tag{2}$$

where $M \in \mathbb{R}^{H \times W \times 1}$ is the mask matrix generated randomly with the mask ratio $\zeta \in [0, 1)$. Then feature distillation can be formulated as:

$$L_{fd} = \sum_{k=1}^{C^t} \sum_{i=1}^{H} \sum_{j=1}^{W} (F_{i,j,k}^t - F_{i,j,k}^s)^2. \quad (3)$$

Although masked reconstruction achieves promising performance, one potential weakness may affect its gain in semantic segmentation: Reconstructed features tend to be similar in local areas [3]. However, in semantic segmentation, local differences indicate structured knowledge, in which student's reconstructed feature maps should keep these differences. *Here comes out why we choose masked reconstruction as our basic feature distillation method: (i) choosing a stronger baseline to further test the effectiveness of our Af-DCD; (ii) examining whether our Af-DCD can promote masked reconstruction for generating more structured representations, enhancing feature imitation process.*

**Motivation and Key Ideas of Af-DCD.** In semantic segmentation, we notice two important facts. On one side, different pixels within local areas may contain different semantic information, as they may belong to different categories or different parts of an object. We call this as *contextual information*. On the other side, differences among channel groups of a pixel representation implicitly indicate the semantic meaning of this pixel, as each output channel is a specific projection of all input feature maps. We call this as *positional channel-group information*. In short, the contextual and positional channel-group information within each local area in spatial and channel dimensions is of significance in semantic segmentation. If both types of information from teacher's feature maps can be densely utilized as the distillation guidance to facilitate the training of the student model, we conjecture that student's performance and generalization capability can be greatly boosted. However, traditional feature imitation loss $L_{fd}$ is difficult to capture these two types of information, as they are not directly observable and measurable comparing to salient representation. Aiming to tackle this problem, we define contrastive loss across student's and teacher's pixel-level or more fine-grained representations to explicitly model such kind of knowledge transfer. Specifically, as shown in Figure 2, Af-DCD incorporates three key ideas:

(1) **Spatial Contrasting.** With the first phenomenon, we leverage pixel-wise representations in $F^s$ and $F^t$ and define pixel-wise dense contrasting based on spatial positions, aiming to transfer teacher's spatial contextual information to student;

(2) **Channel Contrasting.** With the second phenomenon, we then move further on pixel-wise dense contrasting and split every pixel-specific representation into disjoint groups and define group-wise dense contrasting based on channel positions, aiming to force student to learn teacher's positional channel-group information;

(3) **Omni-Contrasting.** Progressively, we unify the above two contrasting methods in an hybrid design, which takes advantage of tactful feature partitions across both channel and spatial dimensions in local areas by splitting the feature map into disjoint patches of the same size.

**Formulation of Af-DCD.** The formulation of Af-DCD follows the notations in Section 3.1. The projected feature maps of teacher and student are denoted as $F^t \in \mathbb{R}^{H \times W \times C^t}$ and $F^s \in \mathbb{R}^{H \times W \times C^t}$, respectively. In the perspective of contrastive learning, we first clarify some important concepts in our Af-DCD:

(1) **Samples and Views.** Different from existing contrastive distillation methods in classification and detection, the samples in Af-DCD are pixel-level or more fine-grained representations in feature maps of the same image, which are naturally dense. Furthermore, we define that representations of teacher and student are two views which should be aligned by our contrastive loss;

(2) **Positive Pairs and Negative Pairs.** Under the definitions above, a positive pair can be defined as: a teacher-student sample pair $(F_i^s, F_j^t)$ which has the same index $i = j$, where $i$ and $j$ are general location indices. Similarly, a negative pair can be defined as: a teacher-student sample pair $(F_i^s, F_j^t)$ whose indices are different $i \neq j$.

---

[3]Experimental results illustrate that the distances between feature partitions within local areas are usually small (see in Figure 3b).

Based on the above definitions, the loss for a specific sample $F_i^s \in F^s$ can be formulated as:

$$l_{Af-DCD}(F_i^s, F^t) = -\log \frac{\exp(-d(F_i^s, F_i^t)/\tau)}{\sum_{j=1}^N \mathbb{1}_{i \neq j} \exp(-d(F_i^s, F_j^t)/\tau)}, \quad (4)$$

where $\tau$ denotes the temperature score, $N$ is the total number of contrastive pairs and $\mathbb{1}_{i \neq j}$ is an indicator function which is 1 iff. $i \neq j$. Different from traditional contrastive loss that adopts cosine distance on measuring similarities, we use Euclidean distance to measure sample pair similarities, where $d(F_i^s, F_i^t) = ||F_i^s - F_i^t||_2^2$. Our ablative experiments illustrate that improved performance would be attained when we select the same type of the distance function for $L_{fd}$ and $L_{Af-DCD}$.

*Spatial Contrasting.* As shown in Figure 2 (top left), aiming to learn teacher's spatial contextual information, Spatial Contrasting constructs dense contrasting operations between teacher's and student's pixel-level representations, where for each sample $F_{i,j}^s$, the positive pair is $(F_{i,j}^s, F_{i,j}^t)$, while the other $\{(F_{i,j}^s, F_{u,v}^t)\}_{u,v \neq i,j}$ are all negative pairs. In such condition, we obtain 1 postive pair and $HW - 1$ negative pairs. Then Spatial Contrasting can be formulated as:

$$l_{Af-DCD}^{SC}(F_{i,j}^s, F^t) = -\log \frac{\exp(-d(F_{i,j}^s, F_{i,j}^t)/\tau)}{\sum_{u=1}^H \sum_{v=1}^W \mathbb{1}_{u,v \neq i,j} \exp(-d(F_{i,j}^s, F_{u,v}^t)/\tau)}. \quad (5)$$

*Channel Contrasting.* As shown in Figure 2 (top right), Channel Contrasting first splits each pixel representation into $M$ non-overlapping channel groups of the same length. Specifically, $F_{i,j}^s$ and $F_{i,j}^t$ are split into $\{F_{i,j,w}^s\}_{w=1...M}$ and $\{F_{i,j,w}^t\}_{w=1...M}$, where $w$ is the index of channel group. Then, in order to transfer teacher's channel-group information at each pixel position, the dense contrasting happens on each pixel-level representation pair between $\{F_{i,j,w}^s\}_{w=1...M}$ and $\{F_{i,j,w}^t\}_{w=1...M}$. For each fine-grained representation $F_{i,j,k}^s$, the positive pair is $(F_{i,j,k}^s, F_{i,j,k}^t)$ and negative pairs are $\{(F_{i,j,k}^s, F_{i,j,w}^t)\}_{w \neq k}$, where the number of positive and negative pairs is 1 and $M - 1$, respectively. Then we substitute these pairs into Formula 4, where $w$ denotes the index. Channel contrasting can be formulated as:

$$l_{Af-DCD}^{CC}(F_{i,j,k}^s, F_{i,j}^t) = -\log \frac{\exp(-d(F_{i,j,k}^s, F_{i,j,k}^t)/\tau)}{\sum_{w=1}^M \mathbb{1}_{w \neq k} \exp(-d(F_{i,j,k}^s, F_{i,j,w}^t)/\tau)}. \quad (6)$$

*Omni-Contrasting.* Omni-Contrasting is an neat combination of Channel Contrasting and Spatial Contrasting, shown in Figure 2 (bottom). Different from the design of Spatial Contrasting, Omni-Contrasting does not contrast all positions of feature maps. Instead, in order to exploit local information, Omni-Contrasting groups pixels into a series of local patches and leverages spatial and channel contrasting within each local patch. This local contrasting can force student to learn teacher's spatial contextual information and positional channel-group information simultaneously, which are beneficial for accurately segmenting the boundary and correctly classify those pixels. After that, $F^t$ and $F^s$ are split into $\{F_p^t\}_{p=1...N}$ and $\{F_p^s\}_{p=1...N}$, where $F_p^t, F_p^s \in \mathbb{R}^{\hat{H} \times \hat{W} \times C^t}$ and $N = \frac{HW}{\hat{H}\hat{W}}$. Then, like the operation in Channel Contrasting, the channels will be divided into $M$ non-overlapping groups. Teacher's and student's representations at position $(i, j, k)$ in local patch $p$ are denoted as $F_{p,i,j,k}^t$ and $F_{p,i,j,k}^s$, respectively, where $F_{p,i,j,k}^t, F_{p,i,j,k}^s \in \mathbb{R}^{1 \times 1 \times \frac{C^t}{M}}$. For $F_{p,i,j,k}^s$, the positive pair is $(F_{p,i,j,k}^s, F_{p,i,j,k}^t)$, while negative pairs are $(F_{p,i,j,k}^s, F_{p,u,v,w}^t)$ for any $u, v, w \neq i, j, k$, where the number of positive and negative pairs is 1 and $\hat{H}\hat{W}M - 1$, respectively. The Omni-Contrasting can be defined as:

$$l_{Af-DCD}^{OC}(F_{p,i,j,k}^s, F_p^t) = -\log \frac{\exp(-d(F_{p,i,j,k}^s, F_{p,i,j,k}^t)/\tau)}{\sum_{u=1}^{\hat{H}} \sum_{v=1}^{\hat{W}} \sum_{w=1}^M \mathbb{1}_{u,v,w \neq i,j,k} \exp(-d(F_{p,i,j,k}^s, F_{p,u,v,w}^t)/\tau)}. \quad (7)$$

Finally, the overall Omni-Contrasting loss can be defined as:

$$L_{Af-DCD}^{OC} = \frac{1}{HWM} \sum_{p=1}^N \sum_{i=1}^{\hat{H}} \sum_{j=1}^{\hat{W}} \sum_{k=1}^M l_{Af-DCD}^{OC}(F_{p,i,j,k}^s, F_p^t). \quad (8)$$

Table 1: Performance comparison of Af-DCD and recent state-of-the-art distillation methods on Cityscapes, evaluated with different teacher-student segmentation network pairs. † denotes distillation without $L_{kd}$. We follow the way in CIRKD [9] to calculate FLOPs. Best results are bolded.

(a) The same framework with different backbones

| Method | Params (M) | FLOPs (G) | mIOU (%) Val | mIOU (%) Test |
|---|---|---|---|---|
| T: DeepLabV3-Res101 | 61.1M | 2371.7G | 78.07 | 77.46 |
| S: DeepLabV3-Res18 | | | 74.21 | 73.45 |
| SKD [20] | | | 75.42 | 74.06 |
| IFVD [23] | | | 75.59 | 74.26 |
| CWD [22] | 13.6M | 572.0G | 75.55 | 74.07 |
| CIRKD [9] | | | 76.38 | 75.05 |
| MasKD [17] | | | 77.00 | **75.59** |
| **Af-DCD** | | | **77.03** | 75.12 |
| S: DeepLabV3-MBV2 | | | 73.12 | 72.36 |
| SKD [20] | | | 73.82 | 73.02 |
| IFVD [23] | | | 73.50 | 72.58 |
| CWD [22] | 3.2M | 128.9G | 74.66 | 73.25 |
| CIRKD [9] | | | 75.42 | 74.03 |
| MasKD [17] | | | 75.26 | 74.23 |
| **Af-DCD** | | | **76.38** | **75.06** |

(b) Different frameworks with the same backbone

| Method | Params (M) | FLOPs (G) | Val mIOU (%) |
|---|---|---|---|
| T: PSPNet-Res101 | 68.07M | 1868.5G | 78.34 |
| S: DeepLabV3-Res18 | | | 73.20 |
| SKD [20] | | | 73.87 |
| CWD [22] | | | 75.93 |
| MGD†[15] | 13.6M | 572.0G | 76.02 |
| MGD [15] | | | 76.31 |
| **Af-DCD†** | | | 76.44 |
| **Af-DCD** | | | **76.52** |
| S: PSPNet-Res18 | | | 69.85 |
| SKD [20] | | | 72.70 |
| CWD [22] | | | 73.53 |
| MGD†[15] | 12.9M | 507.4G | 73.63 |
| MGD [15] | | | 74.10 |
| **Af-DCD†** | | | 73.92 |
| **Af-DCD** | | | **74.22** |

# 4 Experiments

## 4.1 Datasets and Experimental Setups.

**Datasets.** Five popular semantic segmentation datasets, including Cityscapes [27], Pascal VOC [28], Camvid [29], ADE20K [30] and COCO-Stuff-164K [31], are used in our experiments. We conduct our ablation studies on both Cityscapes and ADE20K, which help us to ensure the best setting in most experiments and analyse the effectiveness of our design. *Details for these five datasets are described in supplemental material.*

**Experimental Settings.** Following general settings [9, 20, 15] in semantic segmentation distillation, we adopt DeeplabV3 [32] and PSPNet [3] for segmentation framework, ResNet-18 [33] and Mobilenetv2 [34] for student backbones, Resnet-101 for teacher backbone and group various teacher-student pairs. In order to make fair comparison with different state-of-the-art methods, we implement our method on both MMSegmentation codebase [35] and CIRKD codebase [9]. In training and evaluation, we use mean Intersection-over-Union (mIoU) to measure the performance of all methods. In training phase, all models are optimized by SGD with the momentum of 0.9, the initial learning rate of 0.02, and the batch size of 16. The input size is $512 \times 1024, 400 \times 400, 512 \times 1024, 512 \times 1024$, for experiments on Pascal VOC, CamVid, ADE20K and COCO-Stuff-164K, respectively. The input size for experiments on Cityscapes are different in the two codebase, $512 \times 1024$ in CIRKD codebase and $512 \times 512$ in MMSegmentation codebase [15]. In evaluation phase, we follow general settings in [22], which evaluate the performance with the original image size. Our masked reconstruction generator consists of two $3 \times 3$ convolutional layers with ReLU, following [15]. *Other default hyper-parameter settings and implementation details are described in supplemental material.*

## 4.2 Main Results

In this part, we intend to compare the distillation performance of our method Af-DCD with recent state-of-the-art methods for semantic segmentation. Aiming for comprehensive comparison, we conduct experiments on the aforementioned five public datasets following general settings.

**Results on Cityscapes.** In Table 1, we conduct experiments on the most popular Cityscapes dataset to validate the generalization ability of our method to different teacher-student network pairs. The experimental results show that Af-DCD outperforms state-of-the-art methods in most cases, with the maximal margin of 1.12%. In average, Af-DCD brings 3.44% gain to the baseline student models, with the maximal gain of 4.37%. From the results shown in Table 1a, we can see that our method can well handle teacher-student network pairs in which students (e.g., DeepLabV3-Res18 and DeepLabV3-MBV2) have the same segmentation framework but with different backbones. The

Table 2: Performance comparison of Af-DCD and recent state-of-the-art distillation methods on the other four datasets. We follow the way in CIRKD [9] to calculate FLOPs. Best results are bolded.

(a) Pascal VOC

| Method | Params (M) | FLOPs (G) | Val mIOU (%) |
|---|---|---|---|
| T: DeepLabV3-Res101 | 61.1M | 1294.6G | 77.67 |
| S: DeepLabV3-Res18 | | | 73.21 |
| SKD [20] | | | 73.51 |
| IFVD [23] | 13.6M | 305.0G | 73.85 |
| CWD [22] | | | 74.02 |
| CIRKD [9] | | | 74.50 |
| **Af-DCD** | | | **76.25** |
| S: PSPNet-Res18 | | | 73.33 |
| SKD [20] | | | 74.07 |
| IFVD [23] | 12.9M | 260.0G | 73.54 |
| CWD [22] | | | 73.99 |
| CIRKD [9] | | | 74.78 |
| **Af-DCD** | | | **76.14** |

(b) Camvid

| Method | Params (M) | FLOPs (G) | Test mIOU (%) |
|---|---|---|---|
| T: DeepLabV3-Res101 | 61.1M | 280.2G | 69.84 |
| S: DeepLabV3-Res18 | | | 66.92 |
| SKD [20] | | | 67.46 |
| IFVD [23] | 13.6M | 61.0G | 67.28 |
| CWD [22] | | | 67.71 |
| CIRKD [9] | | | 68.21 |
| **Af-DCD** | | | **69.27** |
| S: PSPNet-Res18 | | | 66.73 |
| SKD [20] | | | 67.83 |
| IFVD [23] | 12.9M | 45.6G | 67.61 |
| CWD [22] | | | 67.92 |
| CIRKD [9] | | | 68.65 |
| **Af-DCD** | | | **69.48** |

(c) ADE20K

| Method | Params (M) | FLOPs (G) | Val mIOU (%) |
|---|---|---|---|
| T: DeepLabV3-Res101 | 61.1M | 1294.6G | 42.70 |
| S: DeepLabV3-Res18 | | | 33.91 |
| CIRKD [9] | 13.6M | 305.0G | 35.41 |
| **Af-DCD** | | | **36.21** |

(d) COCO-Stuff-164K

| Method | Params (M) | FLOPs (G) | Val mIOU (%) |
|---|---|---|---|
| T: DeepLabV3-Res101 | 61.1M | 1294.6G | 38.71 |
| S: DeepLabV3-Res18 | | | 32.60 |
| CIRKD [9] | 13.6M | 305.0G | 33.11 |
| **Af-DCD** | | | **34.02** |

results of Table 1b further show that our method can also generalize well to teacher-student network pairs in which students (e.g., DeepLabV3-Res18 and PSPNet-Res18) have different segmentation frameworks but with the same backbone.

**Results on Four Other Datasets.** In Table 2, we evaluate the performance of Af-DCD on four other datasets including PASCAL VOC, Camvid, ADE20K and COCO-Stuff-164K to examine the generalization of our design to handle different semantic segmentation tasks. According to the results shown in Table 2a-2d, Af-DCD outperforms state-of-the-art methods by significant margins on these four datasets, with the maximal and average margin of 1.75% and 1.22%, respectively. Furthermore, we can observe that our method consistently shows significant absolute mIOU gains (from 1.42% to 3.04%) to different student models on small-size (Camvid), medium-size (Cityscapes and PASCAL VOC) and large-size (ADE20K and COCO-Stuff-164K) datasets.

### 4.3 Ablation Studies

**Ablation Study on Different Loss Terms.** In our formulation, the overall loss contains three distillation terms, including $L_{kd}$, $L_{fd}$ and $L_{Af-DCD}$. Accordingly, we conduct experiments that independently test the gain from each term to explore the nature of Af-DCD. From the results shown in Table 3, we can get following observations: (i) The accuracy gain of $Baseline + L_{Af-DCD}^{OC}$ is slight on relatively small dataset Cityscapes (0.28% mIOU gain), but it is notably pronounced on much larger dataset ADE20K (1.50% mIOU gain); (ii) Compared to $Baseline + L_{fd}$, $Baseline + L_{Af-DCD}^{OC}$ gets student model with better accuracy on ADE20K dataset, while maintaining almost the same training efficiency; (iii) $Baseline + L_{fd} + L_{Af-DCD}^{OC}$ gets student models with 76.44% mIOU and 36.01% mIOU on Cityscapes dataset and ADE20K dataset, respectively, which are obviously better than both $Baseline + L_{fd}$ and $Baseline + L_{Af-DCD}^{OC}$, showing that two loss terms $L_{fd}$ and $L_{Af-DCD}^{OC}$ are complementary; (vi) $L_{kd}$ can further bring minor extra gains, 0.08% and 0.2%, to $Baseline + L_{fd} + L_{Af-DCD}^{OC}$ on Cityscapes and ADE20K dataset, respectively.

**Ablation Study on Different Designs.** In this study, we conduct a set of experiments on our three basic contrasting designs, namely Channel Contrasting (CC), Spatial Contrasting (SC) and Omni-Contrasting (OC), in order to exploit the gains from different contrasting dimensions and further verify the necessity of our OC. In Table 3a and 3b, we can observe two phenomena:

Table 3: Ablation studies on loss terms and their different combinations. The experiments for Cityscapes and ADE20K are conducted on the first teacher-student network pair in Table 1b and 2c, respectively. The training time is measured on 8 NVIDIA RTX A5000 GPUs with 40000 iterations.

(a) Ablation on Cityscapes

| Method | mIOU (%) | ΔmIOU (%) | $T_{train}$ (h) |
|---|---|---|---|
| *Baseline* | 73.20 | n/a | n/a |
| $+L_{fd}$ | 75.88 | +2.68 | 4.02 |
| $+L^{OC}_{Af-DCD}$ | 73.48 | +0.28 | 4.06 |
| $+L_{fd} + L^{OC}_{Af-DCD}$ | 76.44 | +3.24 | 4.25 |
| $+L_{fd} + L_{kd}$ | 76.04 | +2.84 | 4.05 |
| $+L_{fd} + L_{kd} + L^{OC}_{Af-DCD}$ | 76.52 | +3.32 | 4.27 |
| *Baseline* | 73.20 | n/a | n/a |
| $+L_{fd} + L^{CC}_{Af-DCD}$ | 76.23 | +3.03 | 4.13 |
| $+L_{fd} + L^{SC}_{Af-DCD}$ | 76.26 | +3.06 | 4.18 |
| $+L_{fd} + L^{CC}_{Af-DCD} + L^{SC}_{Af-DCD}$ | 76.33 | +3.13 | 4.29 |
| $+L_{fd} + L^{OC}_{Af-DCD}$ | 76.44 | +3.24 | 4.25 |

(b) Ablation on ADE20K

| Method | mIOU (%) | ΔmIOU (%) | $T_{train}$ (h) |
|---|---|---|---|
| *Baseline* | 33.91 | n/a | n/a |
| $+L_{fd}$ | 34.92 | +1.01 | 4.32 |
| $+L^{OC}_{Af-DCD}$ | 35.41 | +1.50 | 4.35 |
| $+L_{fd} + L^{OC}_{Af-DCD}$ | 36.01 | +2.10 | 4.48 |
| $+L_{fd} + L_{kd}$ | 35.22 | +1.31 | 4.34 |
| $+L_{fd} + L_{kd} + L^{OC}_{Af-DCD}$ | 36.21 | +2.30 | 4.51 |
| *Baseline* | 33.91 | n/a | n/a |
| $+L_{fd} + L^{CC}_{Af-DCD}$ | 35.72 | +1.81 | 4.41 |
| $+L_{fd} + L^{SC}_{Af-DCD}$ | 35.22 | +1.31 | 4.45 |
| $+L_{fd} + L^{CC}_{Af-DCD} + L^{SC}_{Af-DCD}$ | 35.81 | +1.90 | 4.54 |
| $+L_{fd} + L^{OC}_{Af-DCD}$ | 36.01 | +2.10 | 4.48 |

(a) Ablation on $\tau$    (b) Ablation on $\lambda_3$    (c) Ablation on $M$    (d) Ablation on $N$    (e) Ablation on $\zeta$

Figure 3: Ablation studies on major hyper-parameters. In (d), 4* denotes $N = 4$ with $4 \times 4$ max pooling, which can further enhance training efficiency. Best viewed with zoom in.

(i) CC has similar gain to baseline model as SC on Cityscapes, while has much stronger performance on ADE20K, exceeding 0.5% to SC; (ii) The combination of CC and SC performs better than CC and SC, but worse than OC on both datasets (in distillation accuracy and training speed). The above two observations show the superiority of OC, which comes from the neat combination of CC and SC. Specifically, OC groups pixels into a number of disjoint local patches and tactfully leverages CC and SC within each local patch instead of the holistic feature maps to better exploit dense and structured local information for contrastive feature mimicking.

Table 4: Ablation study on the choice of the function $d$ in the Omni-Contrasting loss. The experimental setups are the same to Table 1 (default setting without $L_{kd}$). Best results are bolded.

| Function $d$ in Formula 7 | Dataset | mIOU (%) | ΔmIOU (%) |
|---|---|---|---|
| *Baseline* | | 73.20 | n/a |
| $L1$-normed distance | CityScapes | 75.97 | +2.77 |
| Cosine similarity | | 76.10 | +2.90 |
| $L2$-normed distance | | **76.44** | **+3.24** |
| *Baseline* | | 33.91 | n/a |
| $L1$-normed distance | ADE20K | 35.82 | +1.91 |
| Cosine similarity | | 35.95 | +2.04 |
| $L2$-normed distance | | **36.01** | **+2.10** |

**Ablation Study on Choice of the Function $d$ in the Omni-Contrasting Loss.** We compare Formula 7 of our method with 3 types of the function $d$ including $L_2$-normed distance (our choice), cosine similarity (common choice in contrastive loss) and $L_1$-normed distance. From the results summarized in Table 4, we can see that our method always shows significant mIOU gains to the baseline with all 3 types of the function $d$. Comparatively, our method with $L_2$-normed distance is the best, which supports our intuition that improved performance would be attained by choosing the same type of the function $d$ for the feature distillation loss (Formula 3) and the Omni-Contrasting loss (Formula 7).

**Ablation Study on Training Efficiency.** Our Af-DCD is naturally augmentation-free and memory-buffer-free. We evaluate the efficiency of our design in terms of training time and GPU memory occupation. In Table 3a, we can observe that Af-DCD introduces minor extra training cost, from 4.02 hours to 4.25 hours, which only increases 5.7% training time to feature distillation. From the results shown in Table 5, we can see Af-DCD uses less memory and training time, but achieves better performance, compared to CIRKD.

Table 5: Comparison of training resources used by CIRKD and Af-DCD, under the settings of the first teacher-student network pair in Table 1a. Best results are bolded.

| Method | mIOU (%) | GPU memory (G) | $T_{train}$ (h) |
|---|---|---|---|
| CIRKD | 76.38 | 10.09 | 4.34 |
| Af-DCD | **77.03** | **7.94** | **4.23** |

**Ablation Studies on Major Hyper-parameters.** Recall that our method has five hyper-parameters, we also perform experiments to study them. As shown in Figure 3, the effectiveness of our method is relative stable when values of these hyper-parameters are changing. Notably, we find using *max*

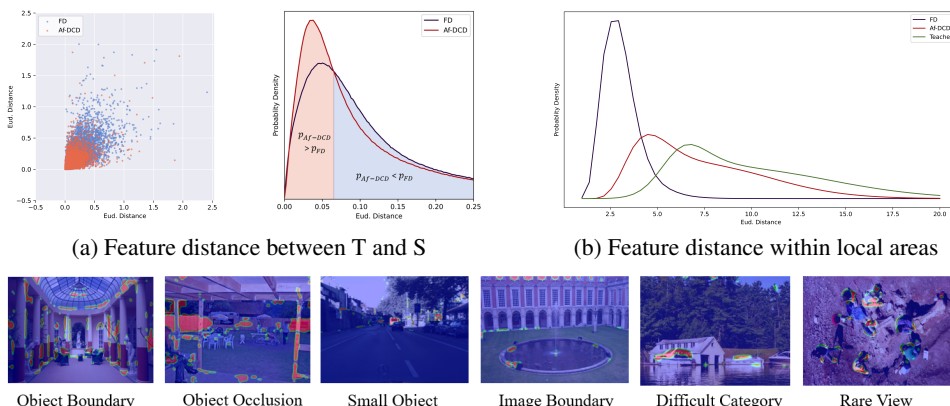

(a) Feature distance between T and S  (b) Feature distance within local areas

(c)  Heat maps of areas where model using Af-DCD segments correctly but using FD segments wrong

Figure 4: Verification experiments. We leverage the models in Table 3b, and count 131M fine-grained representations from 2000 samples. In (a), we randomly select 10K fine-grained representations and measure the distances between student and teacher. In (b), we measure the distances among fine-grained representations within $4 \times 4$ local areas of a specific sample, and randomly select 10K to calculate the probability density. In (c), we choose several difficult cases and highlight areas that $L_{Af-DCD}$ can help $L_{fd}$ segment correctly.

*pooling* with proper scale has no obvious effect on model performance but it has better efficiency than larger $N$. *Other detailed analysis is referred to supplementary materials.*

## 4.4  Discussion

After demonstrating the superior performance of Af-DCD in Section 4.2 and analysing the gain of Af-DCD in Section 4.3, we perform some verification experiments, aiming to discuss how $L_{Af-DCD}$ boosts the distillation performance of $L_{task} + L_{fd}$ (denoted as FD for abbreviation).

As shown in Figure 4b, the self-similarity distribution of fine-grained representations projected by FD has large differences from that of teacher's feature representations. In order to alleviate the aforementioned problem, $L_{Af-DCD}$ contrasts feature partitions within each local patch between student and teacher. The results in Figure 4a and Figure 4b are summarized as below:

(1) $L_{Af-DCD}$ dramatically decreases the distances of student to teacher in views of fine-gained representations and self-similarities, which proves our basic assumption that $L_{Af-DCD}$ can further make student approach teacher in the micro and fine-grained views;

(2) $L_{Af-DCD}$ increases both the mean and variance of the self-similarity distribution of student. Such observation verifies that $L_{Af-DCD}$ can force student to learn dense and structured knowledge implicitly contained among teacher's feature partitions.

The above two improvements are proved to be effective in addressing segmenting various difficult scenarios in semantic segmentation, such as object boundary, small object, object occlusion, difficult category and rare view. Heat maps shown in Figure 4c illustrate that $L_{Af-DCD}$ can effectively help FD correctly classify these difficult pixels, which further verifies the superiority of Af-DCD.

*More examples and detailed analysis are referred to supplementary material.*

## 5  Conclusion

In this paper, we present Af-DCD, an augmentation-free dense contrasive distillation method tailored to semantic segmentation. The dense contrasting in Af-DCD is an omni-dimensional design. Thus, it can effectively transfer teacher's contextual and positional channel-group information to student. Experimental results show that Af-DCD is effective on different teacher-student network pairs and datasets while maintaining training efficiency, as it is born with augmentation-free and memory-buffer free. We hope our work can inspire researchers to explore more powerful and efficient contrastive distillation methods in the future.

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
