# Supplementary Material for "Augmentation-Free Dense Contrastive Knowledge Distillation for Efficient Semantic Segmentation"

**Jiawei Fan**[*]
Intel Labs China
jiawei.fan@intel.com

**Chao Li**
Intel Labs China
chao3.li@intel.com

**Xiaolong Liu**
HoloMatic Technology Co. Ltd.
liuxiaolong@holomatic.com

**Meina Song**
BUPT
mnsong@bupt.edu.cn

**Anbang Yao**[†]
Intel Labs China
anbang.yao@intel.com

## A  Datasets and Experimental Details

### A.1  Datasets

We evaluate our Af-DCD method on five mainstream semantic segmentation datasets following standard training/validation/test splits.

**Cityscapes** [1] is a dataset for real-world semantic urban scene understanding. It has 5,000 image samples with high quality pixel-level annotations and 20,000 image samples with coarse annotations collected from 50 different cities. In semantic segmentation, only samples with pixel-level annotations are used, which contain 2,975 training samples, 500 validation samples and 1,525 testing samples, with totally 19 classes.

**Pascal VOC** [2] is a competition dataset, whose samples are collected from the flickr2 photo-sharing website. In semantic segmentation, the dataset has 21 categories, containing 20 foreground classes and 1 background class, which consists of 10,582 training samples, 1,499 validation samples and 1,456 testing samples.

**CamVid** [3] is a dataset established by Cambridge University in 2008, which can be used in semantic segmentation. It has over 700 samples with pixel-level labels for 11 semantic classes. Specifically, it contains 367 training samples, 101 validation samples and 233 testing samples,

**ADE20K** [4] is a popular semantic segmentation dataset. The scenes in this dataset are challenging, as there are 19.5 instances and 10.5 classes per image on average. Overall, the dataset has 150 categories and around 25,000 images. Specifically, it contains 20,210 training samples, 2,000 validation samples and 3,352 testing samples.

**COCO-Stuff-164K** [5] is a dataset augmented from all 164K images in MS COCO 2017 [6] with pixel-level annotations, which contains 172 classes. Specifically, it has 118,000 training samples, 5,000 validation samples and 20,000 testing samples.

---

[*]This work was done when the first two authors were interns at Intel Labs China, supervised by Anbang Yao who proposed the original idea and led the project.

[†]Corresponding author.

37th Conference on Neural Information Processing Systems (NeurIPS 2023).

Table 1: Performance comparison of Af-DCD and recent state-of-the-art distillation methods on Cityscapes with training-from-scratch setting. * denotes that we do not initialize the backbone with ImageNet pre-trained weights. Best result is bolded.

| Teacher | Student* | Methods | Val mIOU (%) |
|---|---|---|---|
| DeepLabV3-Res101 (78.07) | DeepLabV3-Res18 (65.37) | SKD [7] | 67.08 |
| DeepLabV3-Res101 (78.07) | DeepLabV3-Res18 (65.37) | IFVD [8] | 65.96 |
| DeepLabV3-Res101 (78.07) | DeepLabV3-Res18 (65.37) | CWD [9] | 67.74 |
| DeepLabV3-Res101 (78.07) | DeepLabV3-Res18 (65.37) | CIRKD [10] | 68.18 |
| DeepLabV3-Res101 (78.07) | DeepLabV3-Res18 (65.37) | MasKD [11] | **73.95** |
| DeepLabV3-Res101 (78.07) | DeepLabV3-Res18 (65.37) | Af-DCD (ours) | 73.76 |

## A.2 Experimental Details

**Reference Methods.** We compare our proposed Af-DCD with recent state-of-the-art knowledge distillation methods for semantic segmentation: SKD [7], IFVD [8] , CWD [9], CIRKD [10], MasKD [11] and MGD [12]. In order to make fair comparison, we use the results provided in official implementations or papers.

**Hyper-parameter Settings of Af-DCD.** According to the results in ablation studies described in Figure 3 of the main paper, we empirically use the same settings of hyper-parameters in all experiments. For omni-contrasting, we set temperature score, number of channel groups and patch size as $\tau = 0.7$, $M = 16$ and $N = 4^*$, respectively. For masked reconstruction, we set mask ratio as $\zeta = 0.75$. Finally, for terms of overall loss, we set the coefficients of $L_{kd}$, $L_{fd}$ and $L_{Af-DCD}$ as $\lambda_1 = 1$, $\lambda_2 = 2e - 5$, $\lambda_3 = 5e - 3$, respectively.

# B More Experimental Results

## B.1 Results on Semantic Segmentation

In our main paper, all experiments are conducted on student models pre-trained on ImageNet dataset. Aiming to further explore the efficacy of Af-DCD in training-from-scratch setting, we conduct experiments with DeepLabV3-Res101→DeepLabV3-Res18 teacher-student pair on Cityscapes dataset. The experimental results are shown in Table 1. On the one hand, our Af-DCD shows promising performance, which brings **8.39%** absolute mIOU gain to baseline student model and outperforms CIRKD by a significant margin of 5.58%. On the other hand, however, there exists minor performance gap 0.19% to MasKD. This is because the mechanism of receptive tokens used in MasKD can leverage extra localization information to strengthen supervision signal, which is effective to training-from-scratch setting.

## B.2 Results on Object Detection

In order to further verify the generalization capability of our Af-DCD to more downstream dense predication tasks besides semantic segmentation, we also conduct experiments on MS COCO object detection dataset [6], which contains 118,000 training samples and 5,000 validation samples, with totally 80 categories. We select FKD [13], CWD [9], FGD [14] and MGD [12] for performance comparison. We choose Cascade Mask RCNN-ResX101 [15] as teacher object detector and Faster RCNN-Res50 [16] as student object detector, following the settings in MGD [12]. According to the results shown in Table 2, Af-DCD achieves the best mAP, showing 0.1% absolute mAP gain to MGD (the best of counterpart methods).

## B.3 Results on Tansformer-based Structures

All our experiments in the main paper are conducted with CNN-based teacher-student network pairs. Thus, we apply Af-DCD to transformer-based models to test the effectiveness of our design. Specifically, we choose SegFormer (MiT-B4 encoder as teacher and MiT-B0 encoder as student) [17], a seminal structure for semantic segmentation. Table 3 shows the results, where Af-DCD only brings 0.31% gain to the baseline. This indicates that the design of Af-DCD cannot easily generalize

Table 2: Performance comparison of Af-DCD and recent state-of-the-art distillation methods on MS COCO object detection dataset. Best results are bolded.

| Teacher | Student | Methods | mAP (%) | $AP_S$ (%) | $AP_M$ (%) | $AP_L$ (%) |
|---|---|---|---|---|---|---|
| Cascade Mask RCNN-ResX101 (47.3) | Faster RCNN-Res50 (38.4) | FKD [13] | 41.5 | 23.5 | 45.0 | 55.3 |
| Cascade Mask RCNN-ResX101 (47.3) | Faster RCNN-Res50 (38.4) | CWD [9] | 41.7 | 23.3 | 45.5 | 55.5 |
| Cascade Mask RCNN-ResX101 (47.3) | Faster RCNN-Res50 (38.4) | FGD [14] | 42.0 | **23.8** | 46.4 | 55.5 |
| Cascade Mask RCNN-ResX101 (47.3) | Faster RCNN-Res50 (38.4) | MGD [12] | 42.1 | 23.7 | 46.4 | 56.1 |
| Cascade Mask RCNN-ResX101 (47.3) | Faster RCNN-Res50 (38.4) | Af-DCD (ours) | **42.2** | 23.5 | **46.5** | **56.1** |

Table 3: Experimental results of our Af-DCD on transformer-based structures.

| Method | mIOU (%) | $\Delta$mIOU (%) |
|---|---|---|
| Teacher: SegFormer-MiT-B4 | 81.23 | n/a |
| Student (baseline): SegFormer-MiT-B0 | 75.58 | n/a |
| Af-DCD (ours) | 75.89 | +0.31 |

to transformer-based structures. This is because Af-DCD exploits dense pixel-wise information within each of local patches via the feature partition across both channel and spatial dimensions for formulating contrastive feature mimicking conditioned on the single image input, but transformer-based structures built upon self-attention modules primarily encode global patch-to-patch feature dependencies, which appear to be in conflict with each other.

## B.4 Experiments on Efficient Designs

As we mentioned in the main paper, Af-DCD leverages dense contrasting, but performs efficiently in distillation. This is because we separate the whole feature map into several disjoint patches (called patch separation) and only formulate the Omni-Contrasting within each single patch. Logically, larger patch size has positive influence on the effectiveness of contrastive learning (broader receptive field to construct more negative pairs), but it incurs more computational burden at the same time. Therefore, we should trade-off these two aspects when implementing Omni-Contrasting or find another approach that preserves the both. In order to achieve the later target, based on the hypothesis that difficult negative pairs exist among salient fine-grained representations, we conduct $4 \times 4$ max pooling operations before separating feature map into disjoint $4 \times 4$ patches. This operation is denoted as $4^*$.

We conduct a series of experiments on patch separation (hyper-parameter $N$) and analyze its influences on FLOPs, GPU memory occupation, training time and performance. From the experimental results shown in Table 4, we can observe the following three phenomena: (i) Patch separation can significantly decrease FLOPs from Af-DCD, which makes omni-contrasting possible to be implemented in semantic segmentation distillation; (ii) Larger patch size may incur more FLOPs, more GPU memory occupation and longer training time, but leads better performance; (iii) Our designed operation $4^*$ reduces FLOPs from $1.0e+2$ G to $6.2e-3$ G, GPU memory from 22.32 GB to 7.69 GB and training time from 21.64 hr to 4.48 hr, but only sacrificing 0.01 % mIOU.

Table 4: Experiments on efficient of Af-DCD designs. All experiments are conducted with the PSPNet-Res101→DeepLabV3-Res18 teacher-student network pair. Patch denotes whether leveraging patch separation operations or not. FLOPs only refer the computational complexity of measuring distances when constructing positive and negative pairs in Af-DCD. GPU Memory is the peak GPU memory occupation by the whole model in training stage.

| Method | Patch | Patch Size ($N$) | | | | FLOPs (G) | GPU Memory (GB) | Training Time (hr) | mIOU (%) |
|---|---|---|---|---|---|---|---|---|---|
| | | 2 | 4 | 16 | 4* | | | | |
| Omni-Contrasting | N | | | | | 1.6e+3 | - | - | - |
| Omni-Contrasting | Y | | | ✓ | | 1.0e+2 | 22.32 | 21.64 | 76.45 |
| Omni-Contrasting | Y | | ✓ | | | 6.4 | 7.74 | 5.03 | 76.37 |
| Omni-Contrasting | Y | ✓ | | | | 1.6 | 7.69 | 4.63 | 76.25 |
| Omni-Contrasting | Y | | | | ✓ | 6.2e-3 | 7.69 | 4.48 | 76.44 |

### B.5 Detailed Ablation Studies on Hyper-parameters

In our main paper, we only analyze the results of ablation studies on hyper-parameters in brief. In this part, we will make detailed analysis on each hyper-parameter. The results are shown in Figure 3 in main paper.

**Ablation Studies on $\tau$ and $\lambda_3$.** These two hyper-parameters influence the strength of contrastive constraints. Our method obtains best results when setting $\lambda_3 = 5e - 3$ and $\tau = 0.07$ , where the temperature coefficient is similar to empirical setting in SimCLR [18].

**Ablation Studies on $M$ and $N$.** These two hyper-parameters affect the density of our onmi-contrasting. We have discussed the selection of $N$ in Section B.4. Basically, we aim to increase $M$ for more fine-grained representations. However, there exists underlying redundancy in teacher's feature maps. Too large $M$ may make these fine-grained representations indistinguishable. The experimental results show that $M = 16$ is a relative proper setting.

**Ablation Study on $\zeta$.** For mask ratio, we can find that the gain increases when increasing mask ratio from 0 to 0.75, but dropping after 0.75. The increasing trend indicates properly increasing the mask ratio can adequately leverage the efficacy of mask reconstruction mechanism, while the decreasing after mask ratio surpassing 0.75 may be resulted from that the generator is difficult to reconstruct the feature when mask ratio is too high.

## C Visualization

### C.1 Qualitative Segmentation Visualization

In order to intuitively analyze the efficacy of Af-DCD, we draw color palette segmentation masks of both our Af-DCD and CIRKD based on DeepLabV3-Res101→DeepLabV3-MBV2 network pair trained on Cityscapes. Figure 1 shows the examples on validation set, which verify that Af-DCD can better help student model to achieve more accurate segmentation results. Compared to CIRKD, Af-DCD can help student model to classify difficult pixels stably (case in the first line), to segment object more completely (cases in the second and third line) and to neglect more irrelevant information (case in the last line).

### C.2 T-SNE Visualization

After comparing the qualities of segmentation, we further explore the gain from Af-DCD in terms of class clustering. Af-DCD would push student to be more similar to teacher in feature imitation process. To explore this, we perform the T-SNE [19] visualization on Cityscapes validation set, using DeepLabV3-Res101→DeepLabV3-MBV2 as a test case. We extract pixel-wise representations from the last layer of both teacher and student models and analyse feature distribution. The results in Figure 2 show that Af-DCD has strong capability to transfer teacher's knowledge to student, where the intra-class distances are effectively decreased.

## D Discussion

In Section 4.4 of our main paper, we already had a discussion on how contrastive term in Af-DCD helps student model to mimic structured dense knowledge from the pre-trained teacher model, which contained three significant points: (i) Af-DCD can further help student to approach teacher in the micro and fine-grained views; (ii) Af-DCD can effectively alleviate the high self-similarity problem of masked reconstruction features; (iii) Af-DCD demonstrates promising capability in tackling difficult scenarios in semantic segmentation. In this part, we will further expand the discussion and provide more examples to illustrate the effectiveness of Af-DCD.

**Fine-grained Feature Distances in Training and Validation Set.** In our main paper, we show teacher-student fine-grained feature distances on training set of Cityscapes, in order to illustrate the effect of Af-DCD on feature imitation process. Figure 3 further shows the result comparison on validation set. We can observe that Af-DCD gets similar results on validation set to training set and the efficacy of Af-DCD on minimizing feature differences is generalized. Yet we also notice that the distribution of Af-DCD on validation set is not as concentrated as that in training set, which indicates

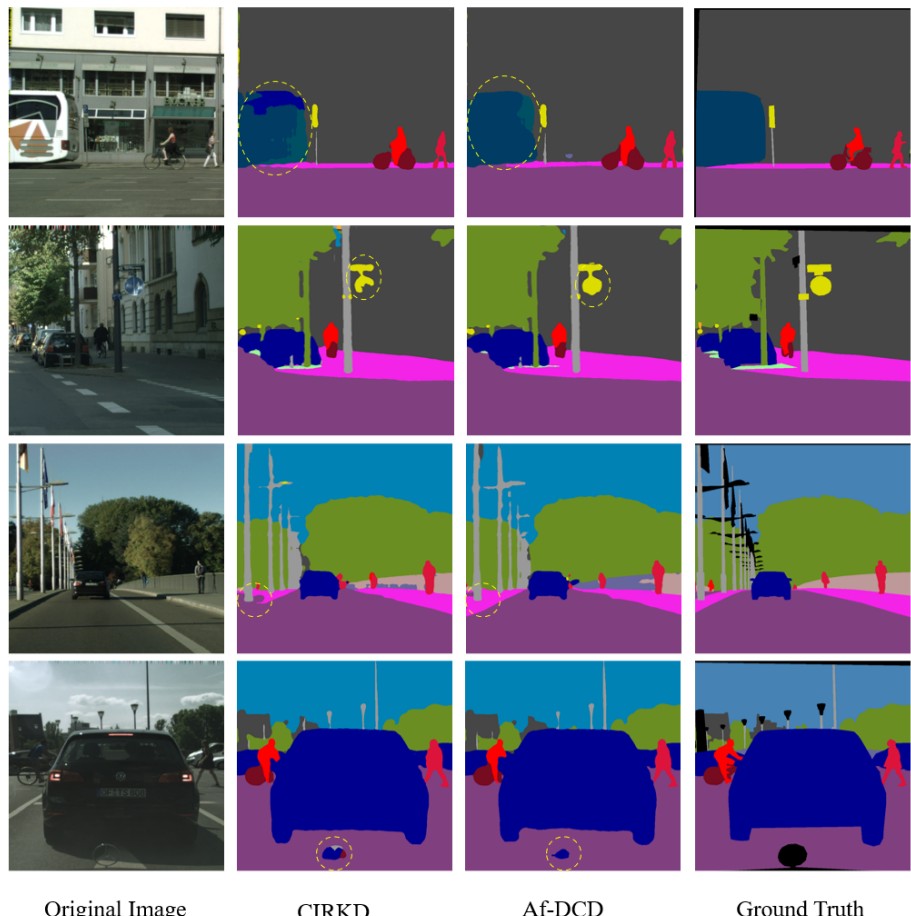

| Original Image | CIRKD | Af-DCD | Ground Truth |

Figure 1: Qualitative segmentation results of DeepLabV3-Res101→DeepLabV3-MBV2 network pair on the validation set of Cityscapes. Comparison among CIRKD (75.42 % mIOU), Af-DCD (76.38 % mIOU) and ground truth. Dash circles denote highlighted areas.

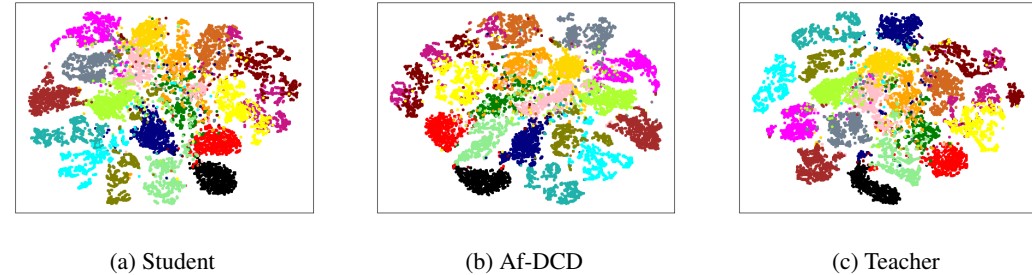

| (a) Student | (b) Af-DCD | (c) Teacher |

Figure 2: Comparison of feature distributions among the individually trained student model, student model trained by Af-DCD and pre-trained teacher model. We use a DeepLabV3-Res101→DeepLabV3-MBV2 network pair well-trained on the Cityscapes dataset, and all images in the validation set. The features are extracted from the last-layer and the feature distributions are analysed by T-SNE [19]. The left figure shows the feature distribution from an individually trained student model; the middle figure shows the feature distribution from the student model trained by Af-DCD; and the right figure shows the feature distribution from the pre-trained teacher model.

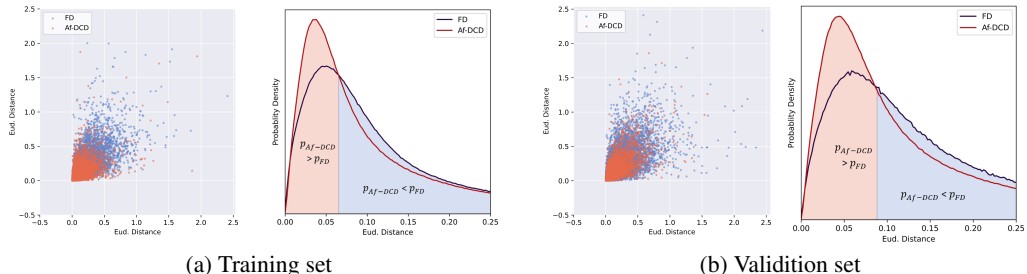

(a) Training set                    (b) Validition set

Figure 3: Fine-grained feature distance distribution between teacher and student on Cityscapes training set (left) and validation set (right). On training set, we obtain 131M fine-grained representation pairs from 2,000 random selected samples. On validation set, we obtain 32.7M fine-grained representation pairs from all 500 samples. Then we calculate distance in each pair. Finally, we randomly select 10,000 distances to draw the scatter and distribution map. Blue points denote masked feature distillation (FD), while orange points denote our Af-DCD.

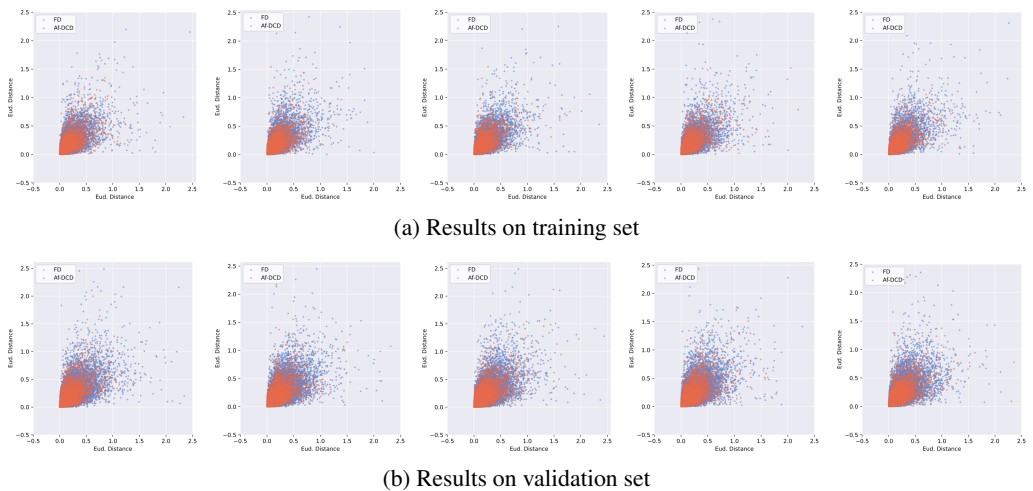

(a) Results on training set

(b) Results on validation set

Figure 4: Sampling results on fine-grained feature distance distribution. The results are randomly sampled from 5 runs on Cityscapes.

the distribution gap between training set and validation set slightly affects the effectiveness of our explicit contrasting among fine-grained representations.

**More Cases on Difficult Scenarios.** Due to limited page space, in the main paper we only place several examples of difficult scenarios. Here in Figure 5, we provide more examples from Cityscapes and ADE20K, which contain 6 types, including object boundary, object occlusion, small object, image boundary, difficult category and rare view. The highlighted areas denote pixels that our Af-DCD can correctly segment, while FD cannot.

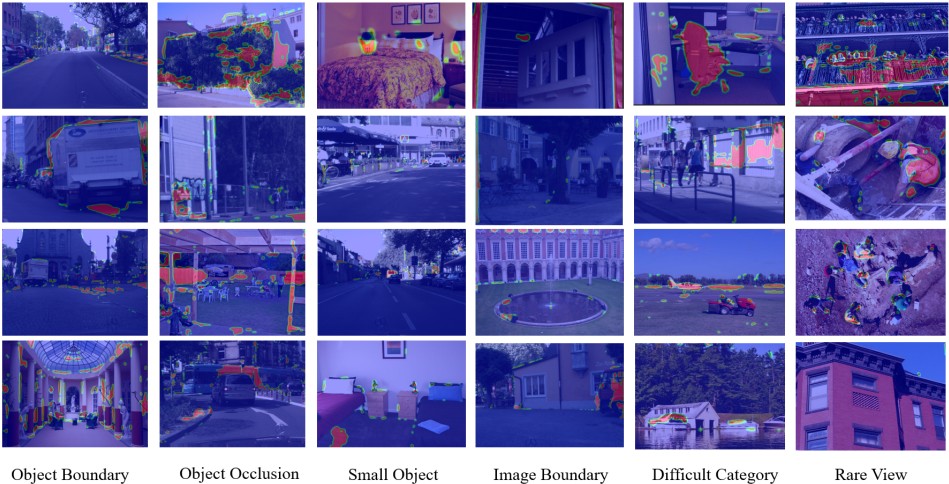

| Object Boundary | Object Occlusion | Small Object | Image Boundary | Difficult Category | Rare View |

Figure 5: More examples of heat maps on segmentation improvements from Af-DCD. The examples are from Cityscapes and ADE20K. Highlighted areas denote that our Af-DCD can correctly segment, while feature distillation method cannot. The models are chosen from Table 4(b) in our main paper.

## E   Limitations of Af-DCD

Although our Af-DCD shows promising performance on various datasets and teacher-student network pairs, it still has limitations. Our preliminary experiments in Table 3 show that the gain of Af-DCD on transformer-based architectures is not as large as on CNN-based architectures. We analyse the reason for this phenomenon is that our Af-DCD cannot effectively leverage global information implicitly contained in pixel-wise representations, as self-attention operation provides global receptive field to representations of all positions. In order to adapt this property, we are planning to make proper modifications on our contrasting paradigm, which zoom out from dense local contrasting into intermediate-level contrasting, neglecting detailed differences among neighbourhood pixels but concentrating more on semantic meaning in local areas. The aforementioned studies will be elaborated in our future research.