# OpenReview forum: "Augmentation-Free Dense Contrastive Knowledge Distillation for Efficient Semantic Segmentation"
_NeurIPS.cc/2023/Conference — NeurIPS 2023 poster_

### Official Review · Reviewer_wLTq · 2023-06-27

**Soundness:** 3 good
**Presentation:** 3 good
**Contribution:** 3 good
**Rating:** 6
**Confidence:** 4

**Summary:**

In this manuscript, the authors propose a effective knowledge distillation framework for semantic segmentation task. Specifically, in addition to traditional knowledge distillation on segmentation masks as well as feature distillation, to better align the dense feature, the authors introduce contrastive learning on both spatial dimension and channel dimension. The proposed Af-DCD loss significantly improves the performance of CNN-based segmentation network via knowledge distillation without data augmentation.

**Strengths:**

1. The motivation is clear, i.e., traditional feature distillation loss is somewhat difficult to contextual information and positional channel-group information.

2. The extensive experiments demonstrate the effectiveness of the proposed method. The ablation study and discussion is abundant and valuable.

3. The proposed method is easy to follow.

**Weaknesses:**

1. Intuitively, the proposed Af-DCD loss can align the dense feature independently, therefore the authors could show the ablation study of baseline+L_{Af-DCD} only and compare the results with baseline+L_{fd}.

**Questions:**

See the weakness section.

**Limitations:**

The authors have listed the limitation in the supplementary material, i.e., the gain of Af-DCD on Transformer based architecture are not as large as on CNN based architecture. We believe this limitation can be seen as a future direction.

---

> ### Author Rebuttal · Authors · 2023-08-10
>
> Thank you for the constructive review, and the recognition of our motivation, method and experiments. In what follows, we provide detailed responses to address your concerns one by one:
>
> **1. Your comments on the weakness about the lack of an ablation study** “Intuitively, the proposed Af-DCD loss can align the dense feature independently, therefore the authors could show the ablation study of baseline$+L_{Af-DCD}$ only and compare the results with baseline$+L_{fd}$”.
>
> **Our responses**: **(1)** Yes, our proposed omni-contrasting loss $L_{Af-DCD}^{OC}$ (abbreviated as $L_{Af-DCD}$ in your comments) can align the dense features independently since it is formulated as an effective contrastive distillation learning scheme for transferring dense and structured local knowledge across both channel and spatial dimensions learnt by the pre-trained teacher model to the target student model; **(2)** Following your insightful comments, we perform ablative experiments on Cityscapes and ADE20K datasets using the same experimental setups to Table 4 in the main manuscript. Detailed results are summarized in the below two Tables. It can be seen that **(a)** Baseline$+L_{Af-DCD}^{OC}$ brings significant accuracy gains to baseline models on both Cityscapes and ADE20K datasets; **(b)** Compared to baseline$+L_{fd}$, baseline$+L_{Af-DCD}^{OC}$ gets student models with better accuracy on both Cityscapes and ADE20K datasets, while maintaining almost the same training efficiency; **(c)** The accuracy gain of  baseline$+L_{Af-DCD}^{OC}$ is slight on relatively small dataset Cityscapes (0.10% mIOU gain), but it is notably pronounced on much larger dataset ADE20K (0.49% mIOU gain); **(d)** The ablative experimental results reported in Table 4 of the main manuscript have already shown that baseline$+L_{fd}+L_{Af-DCD}^{OC}$ gets student models with 76.44% mIOU and 36.01% mIOU on Cityscapes dataset and ADE20K dataset respectively, which are obviously better than both baseline$+L_{fd}$ and baseline$+L_{Af-DCD}^{OC}$, showing that two loss terms $L_{fd}$ and $L_{Af-DCD}^{OC}$  are complementary; **(5)** We have appended this ablation study to Table 4. The updated version of Table 4 is referred to Table 1 in the one-page PDF file attached in our top-level responses titled **“Author Rebuttal by Authors”**.
>
> |Method (on Cityscapes)|mIOU (%)|$\Delta$mIOU(%)|$T_{train}(h)$
> |:--|:--:|:--:|:--:|
> Baseline|73.20|n/a|n/a
> +$L_{fd}$|75.88|+2.68|4.02
> +$L_{Af-DCD}^{OC}$|75.98|+2.78|4.06
>
> |Method (on ADE20K)|mIOU (%)|$\Delta$mIOU(%)|$T_{train}(h)$
> |:--|:--:|:--:|:--:|
> Baseline|33.91|n/a|n/a
> +$L_{fd}$|34.92|+1.01|4.32
> +$L_{Af-DCD}^{OC}$|35.41|+1.50|4.35
>
> **2. Your comments on our discussions about the limitations of the proposed method Af-DCD** “The authors have listed the limitation in the supplementary material, i.e., the gain of Af-DCD on Transformer based architecture are not as large as on CNN based architecture. We believe this limitation can be seen as a future direction”.
>
> **Our responses**: Thank you for accepting the limitations of our method we have discussed in the last Section “Limitations of Af-DCD” of the supplementary material. Here, we add more discussions to further improve the clarification on the main limitation of our method: **(1)** The reason for why the current design of our Af-DCD cannot easily generalize to transformer-based structures is: Af-DCD exploits dense pixel-wise information within each of local patches via the feature partition across both channel and spatial dimensions for formulating contrastive feature mimicking conditioned on the single image input fed to both the pre-trained teacher model and the target student model, but transformer-based structures built upon self-attention modules primarily encode global patch-to-patch feature dependencies in an image input, which appear to be in conflict with each other; **(2)** Previously, we performed a distillation experiment to explore this. In the experiment, we applied Af-DCD to SegFormer (MiT-B4 encoder as teacher and MiT-B0 encoder as student) [1], a seminal transformer-based structure for semantic segmentation. Detailed results are summarized in the below table where Af-DCD only brings 0.31% mIOU gain to the baseline; **(3)** A potential direction to address the above issue is how to preserve local information and make a good alignment of Af-DCD to transformed-based structures. Please allow us to leave it as a future research direction.
>
> Method (on Cityscapes)|mIOU(%)|$\Delta$mIOU(%)
> |:--|:--:|:--:|
> Teacher: SegFormer-MiT-B4|81.23|n/a
> Student (baseline): SegFormer-MiT-B0|75.58|n/a
> Af-DCD|75.89|+0.31
>
> [1] Enze Xie, et al. “SegFormer: Simple and efficient design for semantic segmentation with transformers”, NeurIPS 2021.
>
> **Finally**, during the rebuttal phase, we also conducted more experiments to improve ablation studies and added discussions to improve the clarifications of our method. You are referred to our top-level responses titled **“Author Rebuttal by Authors”**, and our responses to the other reviewers for details.
>
> Looking forward to your feedback.

---

> > ### Comment · Reviewer_wLTq · 2023-08-17
> >
> > Thanks for your response. Most of my concern has been solved. I tend to accept this paper.

---

> > > ### Author Response · Authors · 2023-08-18
> > > **Thanks for the Recognition of Our Rebuttal**
> > >
> > > Thank you so much for the recognition of our responses. We are glad to see that you tend to accept our paper.
> > >
> > > We will make more efforts to improve our paper further.
> > >
> > > Many thanks for your constructive comments, time and patience.

---

### Official Review · Reviewer_vSYf · 2023-06-28

**Soundness:** 2 fair
**Presentation:** 3 good
**Contribution:** 2 fair
**Rating:** 5
**Confidence:** 5

**Summary:**

This paper focuses on knowledge distillation for semantic segmentation and introduces an augmentation-free dense contrastive loss function. The student and teacher feature maps are partitioned into patches, and both spatial and channel contrasting is performed within these local neighborhoods. For contrastive loss, positive/negative feature pairs are formed using tether and student features extracted from the same image without using any augmentations. Experiments were conducted on five segmentation datasets and the proposed approach is shown to perform better than various existing works.

**Strengths:**

Paper was easy to follow.
Experiments conducted on several datasets.

**Weaknesses:**

The title and introduction section emphasize "augmentation-free". However, the motivation/need to be "augmentation-free" is not clear to me. In the introduction, the authors claim using augmentations leads to high resource demand which I disagree. The proposed approach passes the same image through teacher and student networks. One could also use the proposed loss function as it is by passing original image to one network and an augmented version of the image to the other network. The computation cost will be almost same except the augmentation operation cost which is usually small compared to the whole forward/backprop cost. In fact, using appropriate augmentations may even be helpful as the model will learn robust features that are invariant to these augmentations.

For most of the pixels (other than those that are close to object boundaries), their neighborhood is surrounded by pixels of the same class, and treating them as negatives in contrastive loss is counter-intuitive to me. Ideally for semantic segmentation, we would want pixels of same class to have similar representations so that they can easily be classified to the same class.

In line 197, authors mentioned that they use euclidean distances instead of cosine similarity in contrastive loss without providing any explanation.

The main contribution of this paper is the contrastive loss function L_{AF-DCD}. All the other loss functions are from prior works. In order to show the effectiveness of this loss function, authors should compare results with and without the proposed loss when all the other loss functions are present, i.e., comparison between (L_kd + L_fd) and (L_kd + L_fd + L_afdkd). Such comparison is not provided in Table (4).

Typo: It should be Table. 3 not 4(a) in line 287.

**Questions:**

Why should the proposed approach be augmentation free? The proposed loss can be used with augmentations also.

Why euclidean distance in contrastive loss?

Current experimental comparisons do not clearly demonstrate the effectiveness of L_{AF-DCD} in the present of all the other losses.

---

> ### Author Rebuttal · Authors · 2023-08-10
>
> Thank you for the constructive review, and the recognition of the proposed approach and the experiments. In what follows, we provide detailed responses to address your concerns one by one:
>
> **1. The first concern about why should the motivation/proposed method be augmentation-free**.
>
> **Our responses**: Our motivation/method to be augmentation-free is mainly due to the contrastive learning formulation for supervised semantic segmentation task instead of unsupervised image classification task: **(1)** Most of existing contrastive learning methods adopt the self-supervised formulation (assigning a single binary label to each image pair) conditioned on heavy data augmentations (DAs) to learn a proper representation for a given backbone from a large amount of unlabeled images, while our method addresses semantic segmentation distillation in which classification needs to be pixel-wise; **(2)** For our task, passing the original image to one of teacher and student networks and its augmented version via e.g., crop-resize, rotation and flip to the other  network (i.e., the way used in existing methods) usually breaks the geometric feature alignment, i.e., features at the same locations are no longer positional same, which leads to conflict with the pixel-wise classification in semantic segmentation distillation. Therefore, our method passes the same image through teacher and student works; **(3)** Our method creates dense negative samples within each local patch of the same image via the partition across channel and spatial dimensions, and thus is augmentation-free and efficient (see Table 3 & 4); **(4)** Yes, our method can be also used with DAs. Actually, for experiments on PASCAL VOC, etc., DAs such as crop-resize and flip are used following common setups in semantic segmentation distillation, but teacher and student networks still share the same augmented image input.
>
> **2. The second concern about why treating the neighborhood pixels of a specific pixel as its negative samples**.
>
> **Our responses**: **(1)** At the first glance, it is indeed counter-intuitive to treat neighborhood pixels of a specific pixel as its negative samples since they usually tend to be the same class. However, in our formulation (formula 1), **we already have the feature imitation loss $L_{fd}$** which directly forces student to be the same as teacher at every pixel for each channel, taking the role to attain your mentioned ideal representation learning goal. **The role of our contrastive loss $L_{Af-DCD}$** is to promote the process of transferring dense and structured local knowledge (which can better classify difficult pixels for object/image boundary, small object, object occlusion, difficult category and rare view, as illustrated in Figure 4(c) of the main Manuscript and Figure 5 of the Appendix) learnt by teacher model to student model. That is, $L_{fd}$ and $L_{Af-DCD}$ conditioned on the same source feature pairs collaboratively work to get improved feature distillation, which is verified by the ablation studies in Table 4(a); **(2)** Existing feature visualization work [1] shows that layer-specific feature channels from a pre-trained CNN model usually have changing salient activations across neighboring locations and channels. In line with it, we also perform an ablation study to compare $L_{Af-DCD}$ and $L_{fd}$. **You are referred to our first set of responses to Reviewer wLTq for details**.
>
> [1] MD Zeiler and R Fergus, "Visualizing and Understanding Convolutional Networks", ECCV 2014.
>
> **3. The third concern about why using Euclidean distance in our contrastive loss**.
>
> **Our responses**: **(1)** The reason is simply due to our intuition that improved performance would be attained by choosing the same type of the function $d$ for the basic feature distillation loss $ L_{fd}$ (formula 3) and the contrastive loss $L_{Af-DCD}^{OC}$ (formula 7) conditioned on the same source features; **(2)** We perform ablative experiments to compare our contrastive loss with 3 types of the function $d$ on Cityscapes and ADE20K datasets. Results show that our method with $L2$-normed distance is the best, which supports the above intuition. **You are referred to our third set of responses to Reviewer YQfC for details**.
>
> **4. The fourth concern about the lack of an ablation study to compare $L_{kd}+L_{fd}$ and $L_{kd}+L_{fd}+L_{Af-DCD}^{OC}$**.
>
> **Our responses**: **(1)** Yes, our core contribution is the augmentation-free contrastive loss function $L_{Af-DCD}^{OC}$ across channel and spatial dimensions (a neat combination of our two basic contributions $L_{Af-DCD}^{CC}$ across channel dimension and $L_{Af-DCD}^{SC}$ across spatial dimension); **(2)** Following your insightful comments, we perform ablative experiments on Cityscapes and ADE20K datasets using the same experimental setups to Table 4. It can be seen that $L_{kd}+L_{fd}+L_{Af-DCD}^{OC}$ performs better than $L_{kd}+L_{fd}$ on both datasets meanwhile maintaining similar training efficiencies, demonstrating the effectiveness of $L_{Af-DCD}^{OC}$ in the presence of all the other losses.
>
> |Method (on Cityscapes)|mIOU (%)|$\Delta$ mIOU(%)|$T_{train}(h)$
> |:--|:--:|:--:|:--:|
> Baseline|73.20|n/a|n/a
> $+L_{kd}+L_{fd}$|76.04|+2.84|4.05
> $+ L_{kd}+L_{fd}+L_{Af-DCD}^{OC}$|76.52|+3.32|4.27
>
> |Method (on ADE20K)|mIOU(%)|$\Delta$ mIOU(%)|$T_{train}(h)$
> |:--|:--:|:--:|:--:|
> Baseline|33.91|n/a|n/a
> $+L_{fd}+L_{kd}$|35.22|+1.31|4.34
> $+ L_{kd}+L_{fd}+ L_{Af-DCD}^{OC}$|36.21|+2.30|4.51
>
> **Finally**, thank you for pointing out the typo in line 287. We will correct it and make a careful job on writing and proofreading to improve the presentation of our final paper. During the rebuttal phase, we also conducted more experiments to improve ablation studies and added discussions to improve the clarifications of our method. You are referred to our top-level responses titled **"Author Rebuttal by Authors”**, and our responses to the other reviewers for details.
>
> Looking forward to your feedback.

---

> > ### Comment · Reviewer_vSYf · 2023-08-18
> > **Thank you for the rebuttal.**
> >
> > Thank you for the rebuttal and the additional ablation studies. The rebuttal address most of my concerns and I increased my rating to 'borderline accept'.

---

> > > ### Author Response · Authors · 2023-08-18
> > > **Thanks for the Recognition of Our Rebuttal**
> > >
> > > Thank you so much for the recognition of our responses. We are glad to see that you have raised your score.
> > >
> > > We will continue to improve experimental comparisons, discussions, and etc., so as to further improve our paper during the final paper revision.
> > >
> > > Many thanks for your constructive comments, time and patience.

---

### Official Review · Reviewer_cEUs · 2023-07-05

**Soundness:** 2 fair
**Presentation:** 2 fair
**Contribution:** 2 fair
**Rating:** 6
**Confidence:** 3

**Summary:**

This paper points out that existing knowledge distillation methods have been heavily relying on tdata augmentation and memory buffer, which require high computational resources and this is further amplified when it comes to segmentation task that requires relatively higher resolutions of feature maps for processing. To alleviate this complexity, the method called Af-DCD is proposed, which aims to tackle segmentation task by leveraging knowledge distillation based on a novel contrastive learning. More specifically, this method first leverages masked feature mimicking strategy and proposes a novel contrastive learning loss. Experimental results confirm that the proposed method is effective.

**Strengths:**

1. This paper is easy to read and understand.

2. The proposed method achieves the best performance against competitors.

3. Numerous discussions and ablations are presented to validate the choices.

**Weaknesses:**

1. In Table 1 and 2, it seems like ours refers to cumulatively adding all the different methods (SKD, IFVD, CWD and etcs..). The presentation needs improvements.

2. It would be better if the authors cite each methods in Table 1 and 2 (SKD, IFVD..) so that the readers do not have to look up what those abbreviations refer to.

3. In section 4.2, it is only 'stated' that the proposed method performs the best. I don't find any analysis, explanations or reasonings. Moreover, Although FLOPs and Params are also included, there is no texts covering them.

4. In line with section 4.2, section 4.3 and 4.4 also lack explanations or reasoning. At least it is not sufficient. These sections are simply stating what the table or figure shows without sufficient analysis or attempts to deliver insights.

**Questions:**

See weaknesses above.

**Limitations:**

Limitations are propoerly addressed in Section E.

---

> ### Author Rebuttal · Authors · 2023-08-10
>
> Thank you for the constructive review, and the recognition of the novelty, the effectiveness, and the ablation studies of our work. In what follows, we provide detailed responses to address your concerns one by one:
>
> **1. The first weakness** “In Table 1...ours refers to cumulatively adding all the different methods...needs improvements”.
>
> **Our responses**: **(1)** In Table 1 and Table 2, **“Ours” actually refers to applying our proposed method Af-DCD independently, but not** cumulatively adding all the different methods (SKD, IFVD, CWD and etc.); **(2)** The symbol “**+**” in each row of Table 1 and Table 2 denotes: applying a specific method independently to the target teacher-student network pair for semantic segmentation on the given dataset; **(3)** We are sorry for this confusion due to the potential misunderstanding of the symbol “**+**”. Following your careful comments, we will remove the symbol “**+**”, clarify the meaning of “Ours”, and improve the presentation of our final paper.
>
> **2. The second weakness** “It would be better if... cite each methods in Table 1...what those abbreviations refer to”.
>
> **Our responses**: Thanks a lot for your great suggestion. Accordingly, we will add references to all methods (SKD, IFVD, CWD, CRIKD, MasKD, MGD, etc.) compared in Table 1, Table 2 and other related Tables.
>
> **3. The third weakness** “In section 4.2...I don't find any analysis... Although FLOPs and Params...no texts covering them”.
>
> **Our responses**: We really appreciate your insightful comments. **(1)** In Section “**4.2 Main Results**”, we intend to compare the distillation performance of our method Af-DCD with recent state-of-the-art methods for semantic segmentation. Aiming for a comprehensive comparison, we conduct a lot of experiments on public datasets following general settings in semantic segmentation distillation: **(i)** We first conduct experiments on the most popular Cityscapes dataset to **validate the generalization ability of our method to different types of teacher-student network pair**. From the results shown in Table 1(a), we can see that our method can well handle teacher-student network pairs in which students (e.g., DeepLabV3-Res18 and DeepLabV3-MBV2) have the same segmentation framework but with different backbones. The results of Table 1(b) further show that our method can also generalize well to teacher-student network pairs in which students (e.g., DeepLabV3-Res18 and PSPNet-Res18) have different segmentation frameworks but with the same backbone; **(ii)** Next, we conduct experiments on four other datasets including PASCAL VOC, Camvid, ADE20K and COCO-Stuff-164K to **validate the generalization ability of our method to various semantic segmentation tasks**. From the results shown in Table 2(a)-(d), we can see that our method consistently shows significant absolute mIOU gains (1.42%~3.04%) to different student models on small-size (Camvid), medium-size (Cityscapes and PASCAL VOC) and large-size (ADE20K and COCO-Stuff-164K) datasets; **(2)** The superior performance of our method to existing methods demonstrates the effectiveness of the proposed omni-contrasting distillation learning scheme which transfers dense and structured local knowledge across both channel and spatial dimensions learnt by the pre-trained teacher model to the target student model; **(3)** The basic goal of our work is to leverage a pre-trained high-capacity (large and accurate) teacher model to improve the training of a low-capacity (smaller and less accurate) student model, enabling efficient deployment of semantic segmentation models. Following CIRKD [9], FLOPs and Params are included in Table 1 and Table 2 to compare the computational cost of the teacher and student networks (e.g., at most 19.09$\times$ Params compression and 18.40$\times$ FLOPs compression).
>
> **4. The last weakness** “In line with section 4.2, section 4.3 … without sufficient analysis...to deliver insights”.
>
> **Our responses**: Although we provide some necessary explanations and analysis in Section “**4.3 Ablative Studies**” and “**4.4 Discussion**”, we agree with you that they are still not sufficient. Restricted by limited page length, we put some detailed analysis and explanations in the supplementary material (see Section B-D), as stated in Line 291-292 of Section 4.3 and Line 307 of Section 4.4. Here, we add more explanations to deliver main insights: **(1)** Note that our main contributions are the Augmentation-free Contrastive Losses $L_{Af-DCD}^{CC}$ across channel dimension, $L_{Af-DCD}^{SC}$ across spatial dimension and $L_{Af-DCD}^{OC}$ across both channel and spatial dimensions (a neat combination of $L_{Af-DCD}^{CC}$ and $L_{Af-DCD}^{SC}$, i.e., our core contribution), which could improve the basic feature distillation loss $L_{fd}$ in our formulation while maintaining training efficiency. Ablative results in Table 4(a)-(b) progressively validate their effectiveness by different loss combinations and datasets, and ablative results in Table 3 validate the training efficiency. Ablations in Figure 3 further study the choices of major hyper-parameters, verifying the robustness of our method; **(2)** Figure 4(a)-(b) provide statistical distributions of feature distance between teacher and student models, and heat map visualizations to validate the key insight of our design: $L_{Af-DCD}^{OC}$ can effectively encourage the student model to mimic dense and structured local knowledge learnt by the teacher model; **(3)** Following your insightful comments, we will include more texts to improve explanations and analysis for Section 4.2, 4.3 and 4.4 of our final paper.
>
> **Finally**, during the rebuttal phase, we also conducted more experiments to improve ablation studies and added discussions to improve the clarifications of our method. You are referred to our top-level responses titled **“Author Rebuttal by Authors”**, and our responses to the other reviewers for details.
>
> Looking forward to your feedback.

---

> > ### Author Response · Authors · 2023-08-21
> > **Genuinely Looking Forward to Your Feedback**
> >
> > Dear Reviewer cEUs,
> >
> > Thanks again for your comments and time. As the deadline for the author-reviewer discussion phase is approaching by today, we sincerely hope to hear your feedback to see if our responses solve your concerns.
> >
> > The merits of our work have been consistently recognized by you and all three other reviewers. On the whole, **all your concerns refer to improving the presentation of "Section 4 Experiments"**. To the best of our understanding, we believe that our responses should have cleared your concerns. We genuinely hope you could check our responses, and kindly let us know your valuable feedback. We would be happy to provide any additional clarifications that you may need.
> >
> > Best regards,
> >
> > Authors

---

### Official Review · Reviewer_YQfC · 2023-07-06

**Soundness:** 3 good
**Presentation:** 2 fair
**Contribution:** 3 good
**Rating:** 6
**Confidence:** 4

**Summary:**

This paper proposes a novel knowledge distillation methods for semantic segmentation, called Augmentation-Free Dense Contrastive Knowledge Distillation(Af-DCD). Af-DCD is a new attempt on the usage of contrastive learning in the task of knowledge distillation for semantic segmentation, which alleviate the problem of high computational resource brought by data augmentation and memory buffer. Af-DCD utilizes feature partitions across both channel and spatial dimensions, allowing to effectively transfer dense and structured local knowledge learnt by the teacher model to a target student model while maintaining training efficiency. Experimental results on mainstream benchmarks including demonstrate  the effectiveness of the proposed Af-DCD.

**Strengths:**

1.The experiments are sufficient. A lot of experiments and visual analysis have proved the effectiveness and superior performance of the proposed Af-DCD.
2. The design of Af-DCD is clever and makes use of the structural information of teachers from the aspects of both channel and space.
3. The overall experiment is solid and the code is available, which is nice.


**Weaknesses:**

1. The organization of reference is poor. Reference is not added to specific methods in the table. And there is no reference for MaskKD in the whole paper, which actually refers to [17], which is confusing. The instruction for CKD in section2 actually is the instruction for CWD.
2. Some ablation studies are missing. For example, the lack of combination of Channel Contrasting and Spatial Contrasting in Table 4(a), the choice of function d in formula 7.
3. No distillation experiment of transformer-based structure has been carried out, and it is explained in the appendix that the transformer-based structure gains little improvement from Af-DCD, which limits the generality of Af-DCD.


**Questions:**

1. Is Omni-Contrasting necessary? The combination of Channel Contrasting and Spatial Contrasting in Table 4(a) is needed to demonstrate the superiority of Omni-Contrasting.
2. Please add some ablation studies on the choice of the function d in formula 7, which can not only help to screen the appropriate function but also increase interpretability.

---

> ### Author Rebuttal · Authors · 2023-08-10
>
> Thank you for the constructive review, and the recognition of our work. In what follows, we provide detailed responses to address your concerns one by one:
>
> **1. The first weakness about the organization of reference**.
>
> **Our responses**: **(1)** We agree with you, and will add the references to specific methods in Table 1, Table 2 and other related Tables; **(2)** You are correct, MasKD in Table 1 refers to [17] published in ICLR 2023. MasKD uses a set of learnable embeddings to localize the pixels of interest and generate the distillation masks; **(3)** Yes, in Section 2, [22] should refer to CWD but not SKD which should refer to [20]. We are sorry for these typos in reference; **(4)** Following your careful comments, we will fix confusing/inaccurate references and make a careful job on proofreading to improve the presentation of our final paper.
>
> **2. The second weakness and the first question about the lack of ablation study** “The combination of Channel Contrasting and Spatial Contrasting in Table 4(a)”.
>
> **Our responses**: **(1)** Following your insightful comments, we perform ablative experiments on Cityscapes and ADE20K datasets using the same experimental setups to Table 4. Detailed results are shown in the below two Tables. It can be seen that the combination of Channel Contrasting (CC) and Spatial Contrasting (SC) performs better than CC and SC, but worse than Omni-Contrasting (OC), on both datasets; **(2)** Note that **the concepts of Augmentation-free CC and SC are two basic contributions of our work**. Under this context, the direct combination of them can be viewed as our Vanilla OC. Comparatively, the proposed OC is smarter and neater, which groups pixels into a number of disjoint local patches and tactfully leverages CC and SC within each local patch instead of the holistic feature maps to better exploit dense and structured local information for contrastive feature mimicking, showing the superiority both in distillation accuracy and training speed; **(3)** Now, it is clear that **Omni-Contrasting is indeed necessary**.
>
> |Method (on Cityscapes)|mIOU (%)|$\Delta$mIOU(%)|$T_{train}(h)$
> |:--|:--:|:--:|:--:|
> Baseline|73.20|n/a|n/a
> $+L_{fd}+L_{Af-DCD}^{CC}$|76.23|+3.03|4.13
> $+L_{fd}+L_{Af-DCD}^{SC}$|76.26|+3.06|4.18
> $+L_{fd}+L_{Af-DCD}^{CC}+L_{Af-DCD}^{SC}$|76.33|+3.13|4.29
> $+L_{fd}+L_{Af-DCD}^{OC}$|76.44|+3.24|4.25
>
> |Method (on ADE20K)|mIOU (%)|$\Delta$mIOU(%)|$T_{train}(h)$
> |:--|:--:|:--:|:--:|
> Baseline|33.91|n/a|n/a
> $+L_{fd}+L_{Af-DCD}^{CC}$|35.72|+1.81|4.41
> $+L_{fd}+L_{Af-DCD}^{SC}$|35.22|+1.31|4.45
> $+L_{fd}+L_{Af-DCD}^{CC}+L_{Af-DCD}^{SC}$|35.81|+1.90|4.54
> $+L_{fd}+L_{Af-DCD}^{OC}$|36.01|+2.10|4.48
>
> **3. The second weakness and the second question about the lack of ablation study** “the choice of the function $d$ in formula 7”.
>
> **Our responses**: Following your insightful comments, we perform ablative experiments on Cityscapes and ADE20K datasets using the same experimental setups to Table 4. Specifically, we compare formula 7 of our method with 3 types of the function $d$ including $L2$-normed distance (our choice), cosine similarity (common choice in contrastive learning research) and $L1$-normed distance. Detailed results are summarized in the below two Tables. It can be seen that **(a)** Our method always shows significant mIOU gains to the baseline with all 3 types of the function $d$; **(b)** Comparatively, our method with $L2$-normed distance is the best, which supports our intuition that improved performance would be attained by choosing the same type of the function $d$ for the feature distillation loss (formula 3) and the omni-contrasting loss (formula 7) conditioned on the same source features.
>
> Function $d$ in Formula 7 (on CityScapes)|mIOU(%)|$\Delta$mIOU(%)|
> |:--|:--:|:--:|
> Baseline|73.20|n/a
> $L1$-normed distance|75.97|+2.77
> Cosine similarity|76.10|+2.90
> $L2$-normed distance|76.44|+3.24
>
> Function $d$ in Formula 7 (on ADE20K)|mIOU(%)|$\Delta$mIOU(%)|
> |:--|:--:|:--:|
> Baseline|33.91|n/a
> $L1$-normed distance|35.82|+1.91
> Cosine similarity|35.95|+2.04
> $L2$-normed distance|36.01|+2.10
>
> **4. The third weakness about the limited generality of our method to transformer-based structures**.
>
> **Our responses**: **(1)** The current design of Af-DCD cannot easily generalize to transformer-based structures, as we discussed in the last Section “Limitations of Af-DCD” of the Appendix. **The main reason** is: Af-DCD exploits dense pixel-wise information within each of local patches via the feature partition across both channel and spatial dimensions for formulating contrastive feature mimicking conditioned on the single image input, but transformer-based structures built upon self-attention modules primarily encode global patch-to-patch feature dependencies, which appear to be in conflict with each other; **(2)** Actually, we had a distillation experiment to explore this. In the experiment, we applied Af-DCD to SegFormer (MiT-B4 encoder as teacher and MiT-B0 encoder as student) [1], a seminal transformer-based structure for semantic segmentation. The below table shows the results, where Af-DCD only brings 0.31% gain to the baseline; **(3)** A potential direction to address the above issue is how to preserve local information and make a good alignment of Af-DCD to transformed-based structures. Please allow us to leave it as future research.
>
> Method (on Cityscapes)|mIOU(%)|$\Delta$mIOU(%)
> |:--|:--:|:--:|
> Teacher: SegFormer-MiT-B4|81.23|n/a
> Student (baseline): SegFormer-MiT-B0|75.58|n/a
> Af-DCD|75.89|+0.31
>
> [1] Enze Xie, et al. “SegFormer: Simple and efficient design for semantic segmentation with transformers”, NeurIPS 2021.
>
> **Finally**, during the rebuttal phase, we also conducted more experiments to improve ablation studies and added discussions to improve the clarifications of our method. You are referred to our top-level responses titled **“Author Rebuttal by Authors”**, and our responses to the other reviewers for details.
>
> Looking forward to your feedback.

---

> > ### Comment · Reviewer_YQfC · 2023-08-18
> >
> > I have read the authors' response. The authors "agree with" many of the weaknesses. In this stage, I would like to see more "rebuttal".  Despite the interesting idea of the specific designed unsupervised method for semantic segmentation, this paper need further improvement.  So I tend to slightly decrease my rating.

---

> > > ### Author Response · Authors · 2023-08-18
> > > **Extra Responses to Your Replying to Our Rebuttal**
> > >
> > > We sincerely appreciate your replying to our rebuttal. Among all your mentioned questions and weaknesses, **in the rebuttal we faithfully agree with two weaknesses of them** and also provide our responses to address both: **(a)** the first weakness on improving reference organization (**we did not miss any related paper in our original submission**) has already been well corrected in the rebuttal, as we believe your suggestions/comments are truly helpful; **(b)** the other weakness about the main limitation of our method **was frankly pointed out by ourselves and discussed in our original submission**, and we admit it again in the rebuttal and provide pilot experiments and analysis for a better study of it. **We believe** that all the other your mentioned questions and weaknesses have been well addressed, demonstrating the effectiveness of our method. Considering the above facts, our rebuttal is decent and honest-to-truth (but not arguing-against-truth), to the best of our understanding. We sincerely hope you can consider the aforementioned factors in final rating.
> > >
> > > Looking forward to your reply.

---

> > > > ### Comment · Reviewer_YQfC · 2023-08-21
> > > >
> > > > Thanks for the authors' response. I made my rating score mainly based on the original submission and the rebuttal.  I will keep my score. And I suggest the authors improve the final version.

---

> > > > > ### Author Response · Authors · 2023-08-21
> > > > > **Thanks for the Recognition of Our Work**
> > > > >
> > > > > We are sincerely appreciated that you keep your score and tend to accept our paper.
> > > > >
> > > > > We will carefully revise and improve the final manuscript of our work w.r.t. your suggestions and our responses.
> > > > >
> > > > > Thanks again for your very thorough and constructive comments, time and patience.

---

### Author Rebuttal · Authors · 2023-08-10

Dear Reviewers, Area Chairs, Senior Area Chairs and Program Chairs,

We sincerely thank all four reviewers for their thorough and constructive comments. We are glad that the novelty, basic experiments and performance of our work have been mostly recognized by all four reviewers.

In the past week, we carefully improved the experiments (using all computational resources we have), the clarifications and the discussions of our work to address the concerns, the questions and the requests by all four reviewers. **Summarily, we made the following improvements**:

**(1)** To have a better understanding of the effectiveness of our Af-DCD method, we follow the constructive comments/requests by Reviewer YQfC, Reviewer vSYf and Reviewer wLTq and add several sets of ablative experiments on Cityscapes and ADE20K datasets using the same experimental setups to Table 4, including: **(a)** An ablation study to compare the combination of our basic channel contrasting loss and spatial contrasting loss with our omni-contrasting loss; **(b)** An ablation study to compare our contrasting loss with 3 types of distance function; **(c)** An ablation study to compare the results with and without our omni-contrasting loss in the presence of all non-contrasting losses; **(d)** An ablation study to compare our basic feature imitation loss and our omni-contrasting loss; **(e)** Besides the ablation studies reported in the original manuscript, these new ablation studies further demonstrate the effectiveness of our method, and show more insights by experimental observations.

**(2)** We follow the constructive comments/suggestions/requests from all four reviewers, and add more discussions and clarifications to improve the presentation, the explanations of our design insights, experiments and method’s limitations.

**Finally, in the attached one-page PDF file, all the aforementioned experimental results are summarized in different Tables**. We will include the above experiments and discussions in our final paper. We hope our detailed responses are helpful to address the concerns, the questions and the requests of all four reviewers.

---

> ### Comment · Area_Chair_Q8np · 2023-08-18
>
> Thank the authors for the rebuttal. PCs and I have reminded the reviewers to respond to the rebuttals as soon as possible. The final decision will depend on both the reviews and rebuttal.
>
> @Reviewers: This message is yet another reminder. Please try to respond to the rebuttal asap.
>
> --AC

---

### Decision · Program_Chairs · 2023-09-21

**Decision:**

Accept (poster)

**Comment:**

Four experts reviewed the paper, and all recommended Borderline/Weak Accept. The reviewers liked the method and the ablation studies, and they were impressed by the overall results. The rebuttal did a good job answering the reviewers' questions and/or concerns, so no reviewer recommended rejection after the rebuttal. Hence, the decision is to recommend the paper for acceptance. The authors are encouraged to incorporate the reviewers' suggestions and the rebuttal into the revision.